



# Mapping Antarctic Geothermal Heat Flow with Deep Neural Networks optimized by Particle Swarm Optimization Algorithm

Shaoxia Liu[1,2], Xueyuan Tang[2,3,4,5], Shuhu Yang[1], Lijuan Wang[2,6]

[1] College of Information Technology, Shanghai Ocean University, Shanghai 201306, China;

[2] Key Laboratory of Polar Science of Ministry of Natural Resources (MNR), Polar Research Institute of China,Shanghai 200136, China

[3] School of Oceanography, Shanghai Jiao Tong University, Shanghai 200230, China

[4] Key laboratory of Polar Atmosphere-ocean-ice System for Weather and climate,Ministry of education, department of Atmospheric and Oceanic Sciences & institute of Atmospheric Sciences, Fudan University, Shanghai 200438, China

[5]Ocean College, Zhejiang University, Zhoushan 316021, China

[6] College of Surveying and Geo-Informatics, Tongji University, Shanghai 200092, China

*Correspondence to*: tangxueyuan@pric.org.cn; Tel.: +86-137-6113-9049

**Abstract**: The spatial distribution of geothermal heat flow (GHF) beneath the Antarctic Ice Sheet is a major source of uncertainty in projections of ice sheet dynamics and sea-level rise. Direct measurements are sparse, necessitating robust

modeling approaches. In this study, we developed a neural network framework whose architecture and hyperparameters are optimized using a particle swarm optimization (PSO) algorithm. Trained on a global heat flow compilation and a suite of geophysical datasets, our model generates a new GHF map for the entire continent. The model's accuracy in regions lacking direct measurements was confirmed through training density validation, with prediction errors constrained to within 20%. The resulting map delineates a distinct dichotomy: East Antarctica exhibits predominantly

low GHF values (<60 mW m⁻²) with notable exceptions of high heat flow (>80 mW m⁻²) in the Vostok Subglacial Highlands and Gamburtsev Subglacial Mountains. In contrast, West Antarctica is characterized by widespread high heat flow (>60 mW m⁻²), especially in tectonically active regions like the Transantarctic Mountains and the Amundsen Sea sector. These predictions show agreement when compared with direct borehole measurements. Our work offers a new, robust estimate of Antarctic GHF, providing a critical boundary condition for ice sheet models. We suggest that future

improvements in accuracy and interpretability can be gained by assimilating more high-resolution drilling data and integrating physical constraints into the model framework.

## 1 Introduction

Geothermal heat flow (GHF) refers to the heat energy transferred from Earth's interior to the surface via conduction or convection (Pollack et al., 2013). As an important heat source beneath the Antarctic ice sheet, GHF directly affects the

hydrological system under the ice sheet (Kang et al., 2022). Meanwhile, GHF also serves as a key constraint in ice sheet dynamics modeling, enabling estimates of the melting intensity and its distribution (Obase et al., 2023; Seroussi et al., 2017).



High GHF elevates ice sheet basal temperatures, accelerating basal melting and the formation of basal water, thereby affecting ice sheet movement and stability (Pollard et al., 2005; Wearing et al., 2024; Llubes et al., 2006). In addition, the complex interaction between GHF and climate results in a significant degree of variation in Antarctic ice mass distribution. Therefore,

obtaining accurate GHF data is vital for minimizing the errors in ice flow models and improving the reliability of mass balance predictions of ice sheet (Larour et al., 2012). Further, GHF also constitutes a critical basis for reconstructing Antarctic tectonic history (Mareschal & Jaupart, 2013), and lays a significant factor for understanding the feedback mechanisms produced by Antarctic ice mass loss and predicting sea-level change (DeConto et al., 2016).

However, the sparse and uneven distribution of in situ borehole data for GHF, coupled with the severe climatic challenges of direct measurements in the Antarctic continental interior, presents significant challenges for data acquisition (Fisher et al., 2015). Therefore, current large-scale GHF studies mainly rely on geophysical models to derive its distribution. Conventional approaches fall into two categories: one based on the derivation of geothermal processes, such as decreasing west-to-east heat flow derived from some assumptions of geological conditions (Pollard et al., 2005), crustal and upper-mantle heat flow inferred

from seismic models (Shapiro & Ritzwoller, 2004; Shen et al., 2020; Hazzard & Richard, 2024), and Curie temperature depths estimated using satellite magnetometry and thermal models (Maule et al., 2005; Martos et al., 2017). The other was from statistical methods such as multivariate similarity analysis (Stål et al., 2021), Bayesian inversion of multiple datasets (Lösing et al., 2020) and machine learning (Lösing & Ebbing, 2021). These approaches show consistency at continental scales, with greater GHF beneath the West Antarctic and lower GHF in East Antarctica, but considerable discrepancies in predictions at

regional scales. Specifically, process-based modeling approaches are highly dependent on complex mathematical formulations and a precise understanding of geophysical processes, while single-feature analysis is limited by the choice of variables, especially in extreme environments, where the complexity of deconstructing multiple drivers increases significantly. In contrast, statistical approaches, while versatile, are often inadequate to characterize the geologic processes that regulate the heat flow response, thereby restricting the ability to infer nonlinear correlations from multivariate data. Key geologic controls

may be neglected if approaches are simplistic and do not fully account for numerous drivers. In summary, the complexity and high level of uncertainty in the mechanisms of Antarctic GHF further limit the reliability and validity of established approaches, and modeling Antarctic GHF based on reduced physical parameters faces considerable obstacles. This has led to an urgent need to investigate more innovative alternative approach.

In recent years, deep learning algorithms have shown tremendous potential in the field of Earth sciences due to its high accuracy and capacity to handle complex data. Particularly in polar research, deep learning has been successfully applied to a number of tasks, such as super-resolution reconstruction of Antarctic basal topography (Leong et al., 2020), estimation of ice sheet melting rates (Hu et al., 2021), and identification of subglacial lakes (Xu et al., 2017). Notably, deep neural networks (DNN) have attracted considerable attention for their nonlinear modeling capabilities. Research indicates that this approach not only

rivals traditional physical parameterization methods but exceeds them in certain contexts (Burgard et al., 2023), establishing a



solid basis for accurate predictions. As a result, in data-limited polar regions, neural networks hold promise as a pivotal tool for elucidating complex geothermal structures, improving prediction accuracy and extracting dependable insights from sparse, noisy datasets, thus advancing polar science.

Building on this potential, our study introduces a novel framework to construct a continental-scale GHF map of Antarctica. We employ a neural network whose architecture and hyperparameters are systematically optimized by a particle swarm optimization (PSO) algorithm. A key innovation of our approach is the use of a global heat flow dataset for model training, which leverages a diverse range of geothermal environments to enhance the model's predictive power for the Antarctic domain. This paper details our methodology, including dataset construction and model performance evaluation. We present the

resulting Antarctic GHF distribution, compare it with existing models, and discuss the result's uncertainties and its broader implications for glaciology and solid Earth geophysics.

## 2 Data

### 2.1 Global Heat Flow Dataset

The target variable for this study, GHF, was sourced from the latest global heat flow database released by the International

Heat Flow Commission (IHFC). This comprehensive database compiles approximately 90,000 in-situ measurements, primarily acquired from bedrock drill holes and thermal probes, with each entry accompanied by a quality assessment grade. The raw GHF values in the database exhibit an extremely wide range (from -6,120 to 100,000 mW m$^{-2}$). However, such extreme values are typically considered to be local anomalies associated with non-conductive heat transfer processes (e.g., hydrothermal circulation) or measurement artifacts, and thus lack regional representativeness for continental-scale conductive heat flow

modeling (Bachu, 1988).

To construct a reliable and representative dataset suitable for modeling GHF across Antarctica, we implemented a multi-step preprocessing workflow. First, all marine measurements were excluded to focus on the continental domain, and data points with low-quality assessment grades from the IHFC database were removed. Recognizing that the vast majority of continental GHF values, particularly in Antarctica, fall below 200 mW m$^{-2}$, we employed a custom interquartile range (IQR) method for

outlier detection. By setting the upper and lower bounds at 1.25 and 1.15 times the IQR, respectively, we constrained the dataset to a physically plausible range of 0–200 mW m$^{-2}$. Subsequently, these filtered, high-quality point measurements were aggregated by calculating the mean value within a 0.5° × 0.5° latitude-longitude grid. This gridding procedure consolidates the discrete data points into approximately 10,000 representative grid cells, effectively mitigating point-scale noise and generating a spatially coherent dataset. The final processed GHF dataset has a mean of 65.7 mW m$^{-2}$ and a standard deviation

of 25.6 mW m$^{-2}$ (Fig. 1).



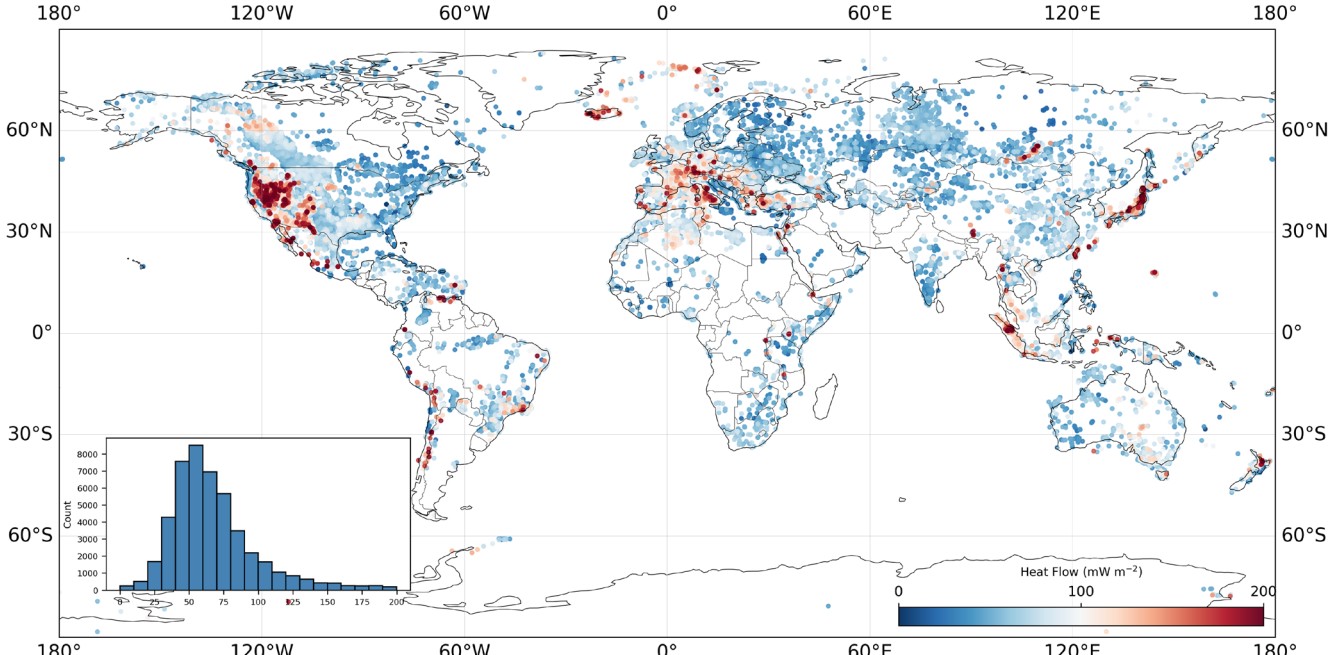

**Figure 1. Spatial distribution of global GHF used to train the model, with colors from blue to red indicating GHF values from low (0 mW m⁻²) to high (200 mW m⁻²) and density histogram of GHF values, where most of the values are concentrated in the range of 50-60 mW m⁻², with a few regions of higher values of 150 mW m⁻² or more. The dataset was obtained from the International Heat Flow Council and the NGHF dataset(Lucazeau,2019), preprocessed by the IQR approach (see Section 2.2 for details).**

## 2.2 Geophysical Features

The spatial distribution of GHF is governed by a complex interplay of the geological and geophysical properties of the lithosphere (Goutorbe et al., 2011; Lucazeau, 2019). To build a robust predictive model, we assembled a comprehensive suite of 16 global-scale feature variables, ensuring complete and consistent spatial coverage that includes the Antarctic continent. These features and their data sources are detailed in Table 1.

**Table 1: Geophysical Features and Sources used for this study.**

| Index | Feature Type | Feature name | Reference |
|:---:|:---:|:---:|:---:|
| 1 | Continuous | Global bedrock topography | ETOPO2022 |
| 2 | | Depth to Moho | Szwillus et al. (2019) |
| | | | An et al. (2015a) |
| 3 | | Lithosphere-asthenosphere boundary | Afonso et al.(2019) |
| | | Depth | Pappa et al. (2019) |



| | | | |
|---|---|---|---|
| 4 | | Thickness of Middle Crust | Laske et al. (2012) |
| 5 | | Thickness of Upper Crust | Laske et al.(2012) |
| 6 | | Pressure Wave Speed | Schaeffer & Lebedev (2015) |
| 7 | | Shear Wave Speed | Schaeffer & Lebedev (2015) |
| 8 | | Bouguer Gravity Anomaly | Bonvalot et al. (2012) Scheinert et al. (2016) |
| 9 | | Curie Temperature Depth | Li et al.(2017) |
| 10 | | Sediment Thickness | Laske et al. (2012) |
| 11 | | Earth Magnetic Anomaly | Maus et al. (2009) |
| 12 | | Gravity Mean Curvature | Ebbing et al.(2018) |
| 13 | Classification | Rock Type | Hartmann & Moosdorf (2012) |
| 14 | | Tectonic Regionalization | Schaeffer & Lebedev (2015) |
| 15 | Proximity | Distance to hot spot | Anderson (2016) |
| 16 | | Distance to Volcanoes | Global Volcanism Program (2013) |

The selected features provide multi-dimensional physical constraints on the thermal state and structure of the lithosphere. Fundamental parameters controlling heat flow include Moho depth, lithosphere-asthenosphere boundary (LAB) depth, crustal thickness, and sediment thickness. Crustal thickness largely determines the total amount of radiogenic heat production from elements such as uranium, thorium, and potassium, which is a primary source of surface heat flow. The LAB depth defines the thermal boundary layer of the lithosphere, with a shallower LAB typically corresponding to a higher geothermal gradient.

Sedimentary layers, due to their low thermal conductivity, act as an insulating blanket, significantly influencing the dissipation of deep-seated heat. Seismic wave velocities, which are inversely correlated with temperature, serve as an effective proxy for the thermal state of the crust and upper mantle. The Curie point depth, corresponding to an isotherm of approximately 580°C, offers a direct constraint on the geothermal gradient. Potential field data, such as Bouguer gravity and magnetic anomalies, indirectly reflect variations in crustal density, composition, and structure, which have empirical relationships with thermal

properties and heat production rates.

To account for the influence of deep mantle processes, we incorporated tectonic and geodynamic features. We utilized the tectonic provinces from the global model of Schaeffer and Lebedev (2015), which is derived from cluster analysis of global surface-wave tomography and has the advantage of not requiring a priori assumptions about Earth's structure. The Global Lithological Map (GLiM) database (Hartmann & Moosdorf, 2012) provides surface rock type data, explaining spatial

variations in thermal conductivity. Furthermore, as active thermal features like volcanoes and hotspots are significant indicators of high advective heat transport, we calculated the distance from the center of each grid cell to the nearest Quaternary volcano and mantle plume hotspot using the Haversine formula. To ensure dataset consistency, all predictor variables were





resampled to a uniform 0.5° × 0.5° grid using Ordinary Kriging. The final feature set thus comprises three data types:
continuous (e.g., crustal thickness), categorical (e.g., lithology, tectonic province), and distance-based (e.g., distance to
volcanoes).

## 3 Methods

Figure 2 illustrates the methodological workflow for modeling GHF across Antarctica. The process begins with the compilation
and preprocessing of the global GHF dataset (the target variable) and the 16 associated geophysical features (as detailed in
Section 2). To ensure model robustness, a collinearity analysis is first performed on the predictor variables to mitigate potential
issues arising from multicollinearity. The core of our methodology is a deep neural network (DNN), whose architecture and
hyperparameters are systematically optimized using a particle swarm optimization (PSO) algorithm. The model's performance
and generalization capability are rigorously evaluated using a 5-fold cross-validation scheme. The final continent-wide GHF
map is generated by ensembling the predictions from the best-performing model in each fold, and the associated model
uncertainty is quantified by the variance among these predictions.


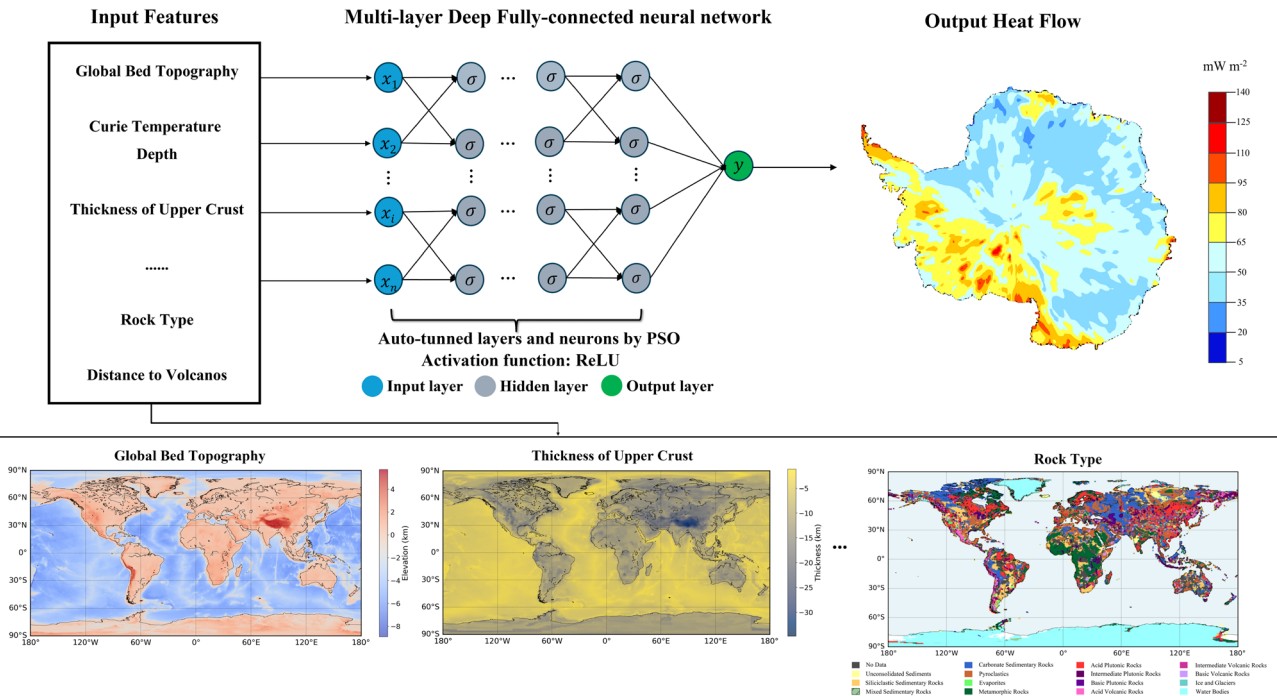

**Figure 2. Schematic of PSO-DNN model structure and GHF prediction. The left side shows the input features of the model. The
middle shows the PSO-DNN model structure, which contains multiple hidden layers (the number of layers and neurons are
determined by the particle swarm optimization algorithm PSO), and ReLU is used as the activation function. The right side shows
the GHF prediction results output by the model, with the color from blue to red indicating the heat flow values from low to high.**



### 3.1 Deep Neural Networks

Neural networks are increasingly utilized in the Earth sciences for their capacity to effectively model complex, non-linear relationships and automatically extract hierarchical features from data (Fausett, 2006; Hastie et al., 2009). This study employs a deep neural network (DNN), an extension of the classic multi-layer perceptron. A DNN consists of an input layer, multiple hidden layers, and an output layer. All hidden layers use the Rectified Linear Unit (ReLU) activation function to mitigate the vanishing gradient problem and enhance computational efficiency. Compared to shallower architectures, increasing the depth with more hidden layers allows the model to learn progressively more abstract and intricate patterns from the input features.

A key advantage of DNNs and other supervised machine learning techniques is their ability to reproduce complex non-linear systems without requiring predefined governing equations. Instead, performance relies on a supervised training phase where the network's internal parameters—the weights and biases of its neurons—are adjusted. During training, the model iteratively tunes these parameters using the backpropagation algorithm, guided by the Adam optimizer, to minimize a mean squared error loss function between the predicted and observed GHF values. The training dataset is randomly partitioned into mini-batches, and the weights are optimized batch by batch. A complete cycle through all mini-batches defines one training epoch, with the weights and biases being continuously refined over multiple epochs. Concurrently, the model's performance on a separate validation dataset is monitored to track its ability to generalize to unseen data. Upon completion of training, the model's final performance is assessed using a test dataset that was entirely withheld from the training and validation processes.

### 3.2 Particle Swarm Optimization

The predictive performance of DNN exhibits high sensitivity to hyperparameter configurations, rendering manual hyperparameter tuning inefficient and suboptimal. To address this challenge, this study employs Particle Swarm Optimization algorithm, a population-based stochastic optimization technique inspired by the collective social behavior of bird flocks (Eberhart & Kennedy, 1995), to systematically search for optimal DNN hyperparameter combinations. In the PSO implementation, each "particle" within the swarm represents a unique candidate set of DNN hyperparameters, encompassing the number of hidden layers, neuron count per layer, initial learning rate, batch size, and regularization strength. The particle swarm iteratively explores the hyperparameter space, with each particle adjusting its trajectory based on its personal best solution and the global best solution to minimize the loss function on the validation set, thereby optimizing both predictive accuracy and generalization capability. The velocity and position updates for particle i follow the equations:

$$v_i(t + 1) = \omega v_i(t) + c_1 r_1(p_i - x_i(t)) + c_2 r_2(g - x_i(t)) \tag{1}$$

$$x_i(t + 1) = x_i(t) + v_i(t + 1) \tag{2}$$





where $v_i(t)$ and $x_i(t)$ represent the velocity and position of the $i$th particle at iteration $t$, respectively, where $p_i$ is the individual optimal position, $g$ is the global optimal position. The inertia coefficient $\omega$ controls momentum preservation, while cognitive

$c_1$ and social $c_2$ coefficients weight the attraction toward $p_i$ and $g$. $r_1$ and $r_2$ are random numbers between [0,1] to provide randomness to enhance the diversity of the search. In this study, PSO was employed to optimize DNN hyperparameters within the following ranges: number of hidden layers (2-8), neurons per hidden layer (12-128), learning rate (0.0001-1.0), batch size (16-100), and regularization strength (0.0001-0.1).

### 3.3 Training process

To satisfy neural network input requirements and optimize training performance, we implemented a two-step preprocessing pipeline for the feature set. First, label encoding was applied to categorical variables such as rock type and tectonic province, converting their non-numerical labels into unique integer representations. Second, all continuous predictor variables and the target GHF variable underwent standardization by subtracting the mean and dividing by the standard deviation, with statistical parameters computed exclusively from training data within each cross-validation fold to prevent data leakage. This

standardization step is crucial for the gradient-based Adam optimizer, ensuring all features operate on similar numerical scales, thereby stabilizing the training process and mitigating risks of slow convergence or gradient explosion.

To robustly evaluate model performance and minimize bias associated with single train-test splits, we employed a 5-fold cross-validation framework. The dataset was partitioned into five mutually exclusive folds, with the model trained five times, each iteration using one fold as the test set and the remaining four as the training set. During each training iteration, the Adam

optimizer was selected to leverage its computational efficiency and adaptive learning rate characteristics. To control model complexity and reduce overfitting risk, L2 regularization was applied, and batch normalization was implemented after each hidden layer to stabilize the learning process and accelerate convergence. Additionally, an early stopping mechanism was established, terminating training if validation loss failed to decrease for 10 consecutive epochs, with model weights corresponding to the lowest validation loss retained.

In the final inference stage, an ensemble model was constructed using the five independent models generated through cross-validation to provide comprehensive coverage across the entire Antarctic continent. The final GHF prediction at any given location represents the arithmetic mean of the five model outputs, with this ensemble strategy enhancing predictive accuracy and robustness by averaging individual model biases. Simultaneously, the standard deviation of the five predictions at each grid point was calculated to serve as a quantitative indicator of model prediction uncertainty.

### 3.4 Model Evaluation Metrics

In order to assess the strength of the model's prediction results, we used two coupling parameters: the coefficient of determination ($R^2$) and the root mean square error (NRMSE) metrics. The combination of these two metrics, combined with a robust procedure to avoid overfitting the input data, provides a widely recognized strategy for assessing the goodness of predictive performance in regression analysis (Branco et al., 2016).






R² is an important measure of the goodness of fit of a model and is used to assess the ability of a model to predict unknown samples. Its value usually ranges from 0 to 1, with the highest value being 1.0, which indicates that the model perfectly explains the variability of the data. The formula for R² is as follows:

$$R^2 = 1 - \frac{\sum_{i=1}^{n}(y_i - \hat{y}_i)^2}{\sum_{i=1}^{n}(y_i - \bar{y})^2} \tag{3}$$


where $y_i$ is the observed value, $\hat{y}_i$ is the predicted value, $\bar{y}$ is the mean of the observed value, and $n$ is the sample size. In this study, the higher R² value indicates that the model is able to effectively capture the relationship between input features and GHF.

NRMSE is a commonly used metric for assessing the relative magnitude of prediction errors, and removes the effect of
magnitude by normalizing the error to a proportion of the predicted mean. The formula is as follows:

$$NRMSE = \frac{\sqrt{\frac{1}{n}\sum_{i=1}^{n}(y_i - \hat{y}_i)^2}}{\bar{\hat{y}}} \tag{4}$$

where $y_i$ is the observed value, $\hat{y}_i$ is the predicted value, $\bar{\hat{y}}$ is the mean of the predicted value, and $n$ is the sample size. In this study, the NRMSE reflects the proportion of prediction error relative to the level of GHF prediction. For example, an error of 0.15 can be interpreted as an average relative error of 15% in the prediction.


## 4 Results

### 4.1 Collinearity Analysis

Collinearity analysis of input features represents a critical step in constructing multivariate regression models, ensuring model stability and interpretability. High linear correlations among predictor variables, known as multicollinearity, inflate the
variance of regression coefficients, thereby compromising predictive performance. Given that GHF is governed by complex interactions among multiple geophysical factors, detecting and mitigating such correlations is essential. To quantify multicollinearity, we employed the Variance Inflation Factor (VIF), which measures the degree to which the variance of regression coefficients increases due to multicollinearity. For the i-th predictor variable, VIF is defined as:

$$VIF_i = \frac{1}{1 - R_i^2} \tag{5}$$


where $R_i^2$ represents the coefficient of determination obtained from regressing the i-th predictor against all other predictor variables.





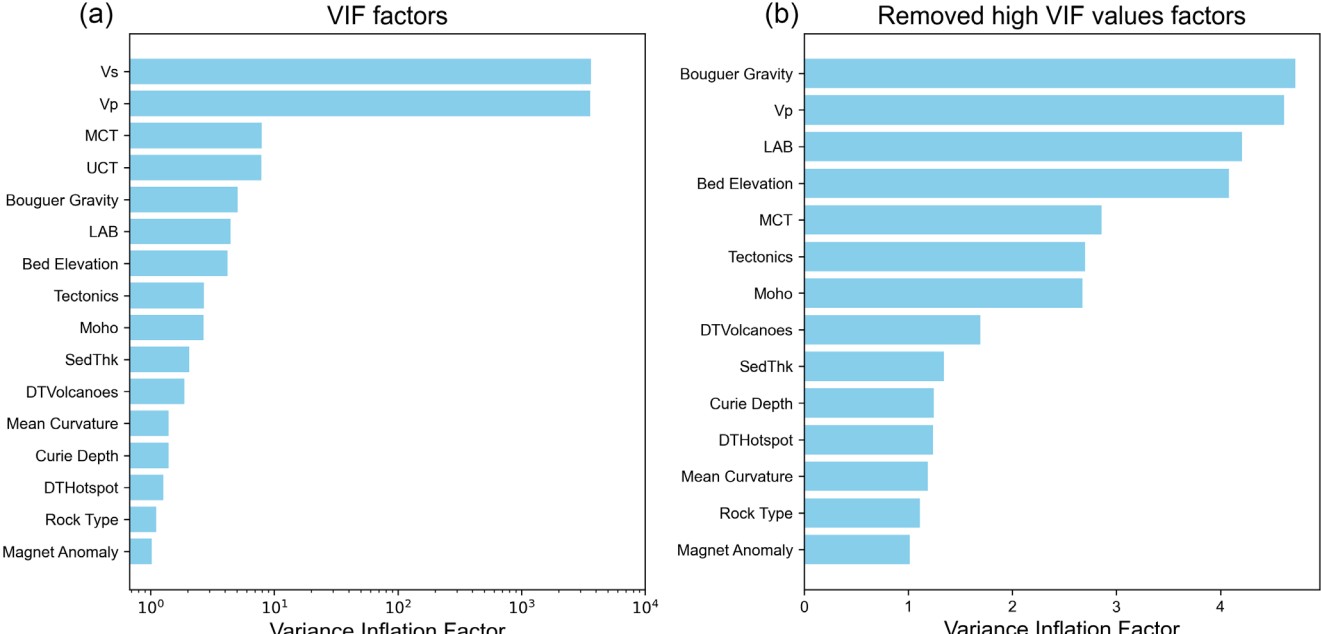

**Figure 3. Collinearity analysis of factors.(a) VIF values for all factors. (b) Collinearity analysis after removing factors with high VIF**
**values. MCT: Middle Crust Thickness; UCT: Upper Crust Thickness;SedThk: Sediment Thickness; DTVolcanoes: Distance to Volcanoes; DTHotspot: Distance to Hotspot.**

The analysis result revealed pronounced collinearity between P-wave velocity and S-wave velocity, as well as between Middle Crust Thickness and Upper Crust Thickness (Fig.3a). Following the removal of Vs and Middle Crust Thickness, VIF values for the remaining features decreased to acceptable levels (Fig.3b). Consequently, these 14 selected features were employed as

inputs for the GHF model, establishing a robust foundation for reliable predictions.

**4.2 PSO Parameter Sensitivity Analysis**

PSO is extensively applied in function optimization and neural network training. The selection of PSO parameters is crucial for algorithm performance and efficiency, as these parameters exhibit interdependencies across different parameter spaces. Typically, parameter selection relies on empirical knowledge. This study employed the pyswarm implementation, with

adjustable parameters including particle number ($m$), inertia weight ($w$), learning factors ($c_1$ and $c_2$), and maximum iterations. The inertia weight controls the influence of a particle's previous velocity on its current trajectory, thereby achieving a balance between global and local search capabilities. We adopted the linearly decreasing weight proposed by Shi & Eberhart (1998):

$$w = w_{max} - \frac{w_{max} - w_{min}}{T_{max}} t \tag{6}$$

where $w_{max}$ and $w_{min}$ represent the maximum and minimum inertia weights (typically set to 0.9 and 0.4, respectively), t denotes the current iteration number, and $T_{max}$ represents the maximum iteration count. The learning factors $c_1$ and $c_2$





determine the stochastic accelerations toward personal best and global best positions, respectively. Previous studies have proposed various recommendations: Kennedy and Eberhart suggested setting both to 2, while subsequent researchers argued for asymmetric values, with experimental evidence supporting $c_1 = 2.8$. Suganthan (1999) tested a method for linearly decreasing both acceleration coefficients over time but observed that fixing acceleration coefficients at 2 produced superior solutions. Regarding particle number, He et al. (2016) demonstrated through their experiments that a particle number of 20 is sufficient for standard optimization problems, whereas more complex scenarios may require up to 50 particles.

Based on prior research, this study designed four experimental configurations with different $c_1$, $c_2$, and m values to determine optimal parameter settings: Config1 ($c_1 = c_2 = 2$, $m = 20$), Config2 ($c_1 = c_2 = 2$, $m = 50$), Config3 ($c_1 = 2.8$, $c_2 = 1.0$, $m = 20$), and Config4 ($c_1 = 2.8$, $c_2 = 1.0$, $m = 50$). The experimental procedure involved PSO-based neural network hyperparameter optimization with the objective of minimizing RMSE on the validation set. Each configuration underwent 100 iterations with linearly decreasing inertia weight while maintaining fixed learning factors and particle numbers. Convergence curves showing RMSE variation with iteration count are presented in Fig. 4.

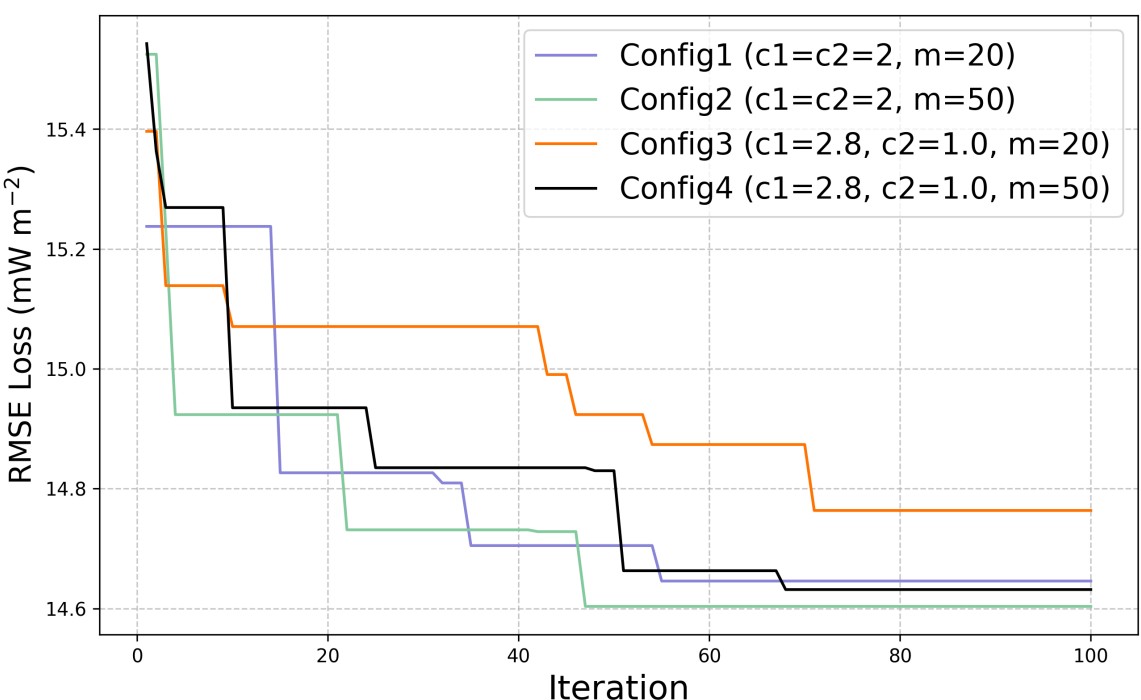

**Figure 4. RMSE Convergence Curves for PSO Configurations. Convergence curves of four PSO configurations for optimizing neural network hyperparameters over 100 iterations.**

Figure 4 illustrates RMSE trends across 100 iterations for the four configurations. Initial RMSE values of approximately 15.4 mW m$^{-2}$ reflect high initial prediction errors. Config1 and Config2 achieved rapid RMSE reduction to approximately 14.6





mW m⁻²before stabilizing, regardless of particle number, indicating that increased particle count did not significantly improve

convergence. Config3 and Config4 demonstrated superior performance, reducing RMSE to 14.8 mW m⁻² within 20 iterations, with further stabilization at 14.6 mW m⁻² when $m = 50$. This indicates that asymmetric learning factor settings combined with higher particle numbers enhance convergence efficiency. Analysis reveals that Config4 maintained the lowest RMSE in later iterations, validating the advantages of asymmetric learning factors and larger particle swarms in complex GHF modeling, thereby providing an optimal parameter foundation for subsequent training.

### 280 4.3 GHF Prediction With Limited Local Data

A significant challenge in this modeling lies in predicting GHF in regions with sparse in-situ measurements, such as Antarctica. To quantitatively assess model performance under such data-constrained conditions and address validation requirements in data-scarce regions, we adopted the training density analysis approach proposed by Rezvanbehbahani et al. (2017). This method systematically evaluates the relationship between prediction accuracy and local training data availability through a

training density metric defined for a specified Region of Interest (ROI):

$$\rho_{ROI} = \left(1 - \frac{N'_{ROI}}{N_{ROI}}\right) \times 100\% \tag{7}$$

where $N_{ROI}$ represents the total number of data points within the target ROI used for training, and $N'_{ROI}$ denotes the number of data points within the ROI that are deliberately excluded from the training set and reserved exclusively for model validation.

This experiment utilized Europe's well-documented dense heat flow dataset as the test subject, with the ROI defined as a representative region covering the most extensive data range. Data points were randomly sampled from the ROI at 10% increments (10% to 90%) and combined with all data points outside the ROI to form the training set. Simultaneously, the remaining data points within the ROI served as an independent validation set to evaluate model prediction performance at corresponding densities. To ensure statistical robustness, this random sampling process was repeated five times at each density

level, with corresponding calculations of mean values and standard deviations for performance metrics.




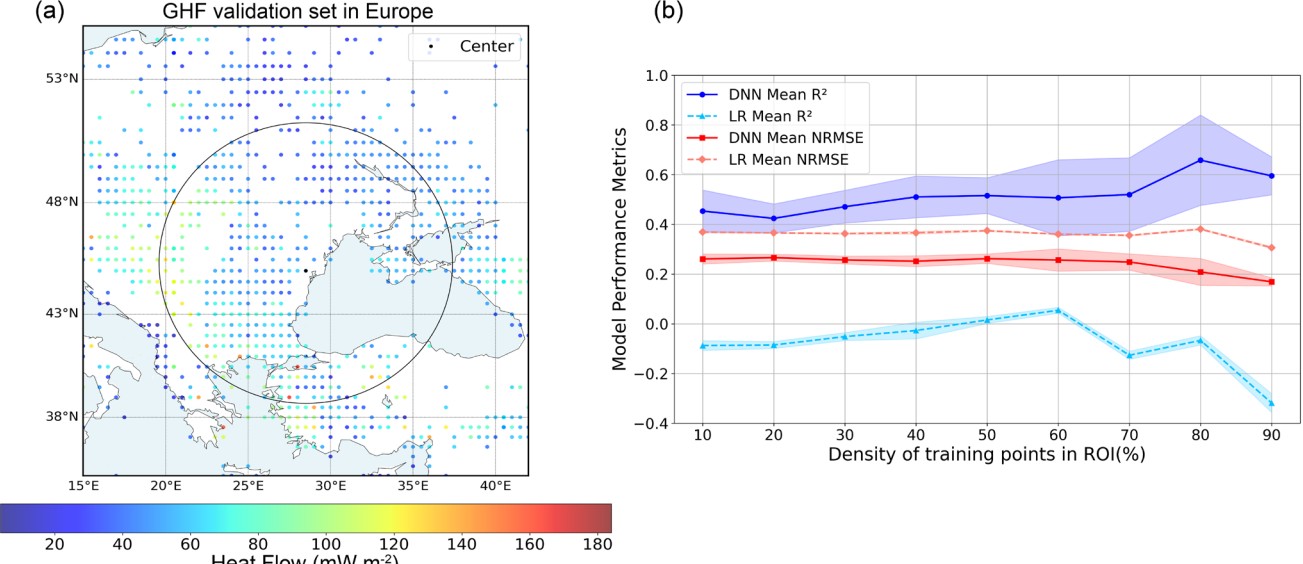

**Figure 5. Performance of DNN and linear regression methods in experiments with different densities of ROI regions. (a). Test region and gray circles represent ROI. (b). Performance of DNN and linear regression under different $\rho_{ROI}$.**

Experimental results(Fig.5) demonstrate a pronounced positive correlation between data density and model performance. As training density within the ROI systematically increased from 10% to 90%, DNN model predictive capability exhibited significant improvement: mean R² values steadily increased from 0.44 to 0.62, while mean NRMSE correspondingly decreased from 0.26 to a minimum of 0.18. In contrast, linear regression achieved R² values progressing from 0.0 to 0.2, with NRMSE remaining within the 0.4-0.5 range.

Analysis reveals significant advantages of DNN over linear regression. Even under 10% local data conditions, DNN achieved an R² of 0.44, demonstrating exceptional robustness attributable to its nonlinear modeling capabilities trained on global datasets, effectively learning geological and geophysical spatiotemporal patterns suitable for regions with insufficient "local experience." Linear regression, constrained by linear relationship assumptions, struggles to capture the complex nonlinear characteristics of GHF, resulting in inferior performance. As training density increases, DNN performance continues to

optimize, validating its iterative improvement capacity for integrating new information, while linear regression remains limited by linear assumptions with constrained improvement potential. These characteristics establish DNN as an ideal tool for GHF prediction in data-scarce regions such as Antarctica.





## 4.4 Antarctic GHF Prediction

We applied our model to the entire Antarctic continent to obtain an integrated anticipated GHF distribution (see Fig. 4). The results suggest that most sections of the East Antarctica have low GHF values ranging from 30-60 mW m⁻². Notably, the lowest GHF values are concentrated in Dronning Maud Land and the Wilkes Subglacial Basin, a characteristic likely associated with the stable Craton lithosphere and limited geothermal activity in these regions. However, in the Gamburtsev Subglacial Mountains, Vostok Subglacial Highlands, and the area around Subglacial Lake Vostok, there is an increasing trend of heat

flow values, which shows that these regions may have been affected by deep tectonic activity or localized heat sources (Artemieva, 2022). In comparison, the heat flow characteristics of the West Antarctica are significantly different, with heat flow values often higher than 60 mW m⁻², indicating more active geothermal activity. The high heat flow values are most prominent along the Transantarctic Mountains belt, and are also broadly distributed along the Amundsen Sea coast, the Siple Coast, and throughout the Antarctic Peninsula region in the West Antarctica. These locations of significant heat flow are tightly

connected with regional tectonic deformation and ice sheet dynamics, exhibiting complicated geologic processes driven by crustal strain, volcanism, or other thermal anomalies.

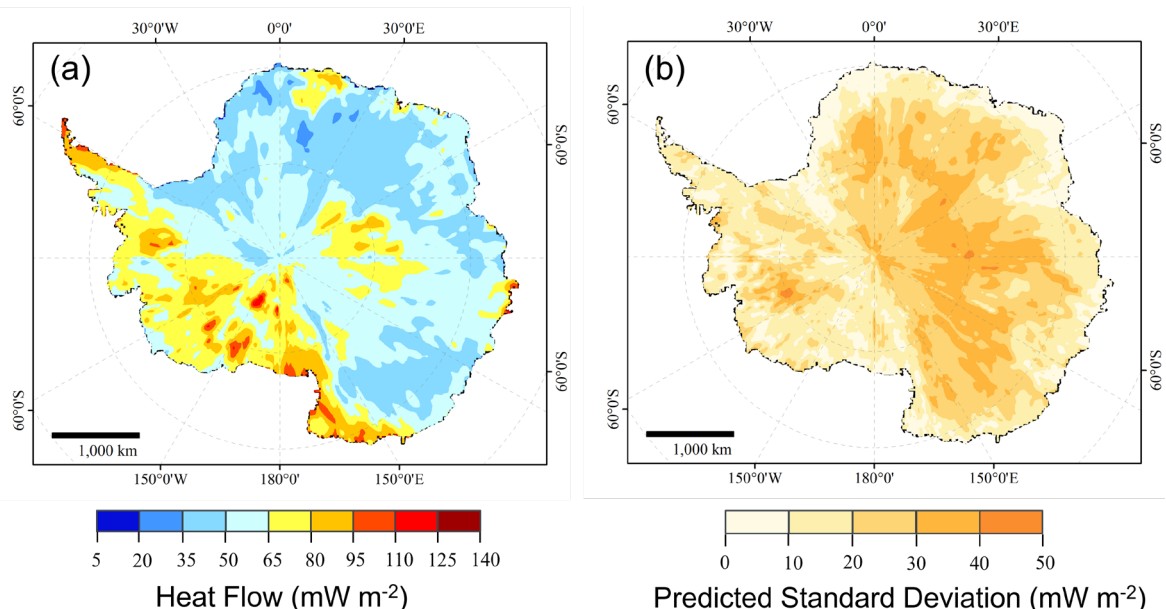

**Figure 6: GHF prediction results and uncertainty analysis of the Antarctic continent. (a) Demonstrates the distribution of GHF predictions across Antarctica, with generally lower heat flow values at East Antarctica and higher values at West Antarctica. (b)**

**Demonstration of the standard deviation of GHF predictions resulting from the five-fold cross-validation.**

Consistent with prior research (An et al., 2015a; Fox Maule et al., 2005; Shapiro & Ritzwoller, 2004), all GHF distribution maps indicate a dichotomous pattern of high values int the west and low in the east of Antarctica (see Fig. 6). This feature is



primarily caused by considerable changes in its tectonic genesis and geological age (Boger, 2011; Veevers, 2012), with active
geology and volcanism considerably influencing regional GHF (Barletta et al., 2018; Haeger et al., 2019). In contrast to previous models, Martos et al. constructed a model based on magnetic data showing higher values of heat flow around West Antarctica (up to 240 mW m⁻²) and lower inland, while Shen et al. used seismic data to show that the GHF is higher along the coast and lower inland, and does not exceed 90 mW m⁻² on a continent-wide scale. Moreover, the model of Losing and Stål et al. falls somewhere in between, presenting a compromise pattern. Our model's prediction results are closest to the distributional
properties of the Martos et al. results (see Fig. 7), but with much lower GHF extremes at the East Antarctica. This mismatch may be mainly related to the fact that very high values in the heat flow data are deleted during data preprocessing to limit the interference of outliers in the model.

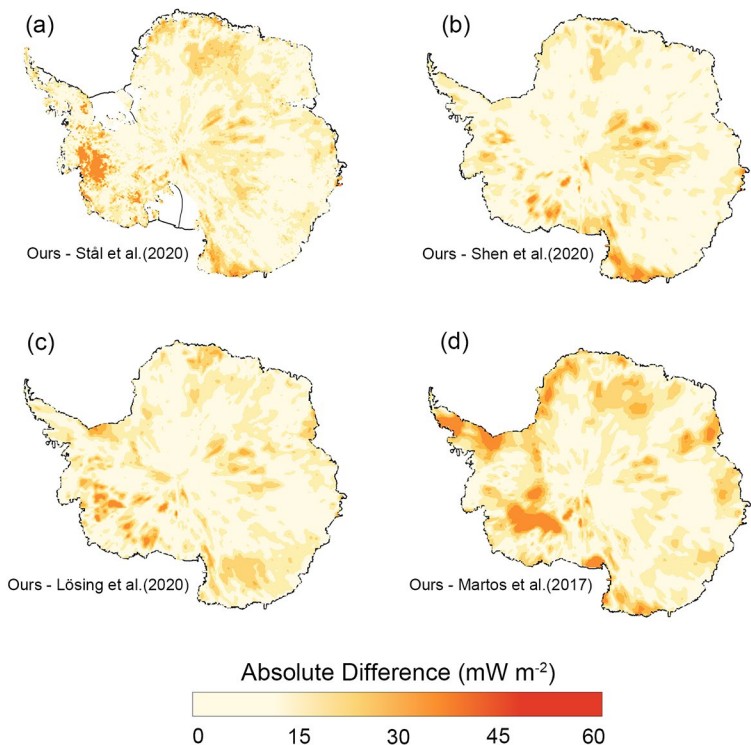


**Figure 7: Comparison of predicted models with previously published models. a) Ours - Stål et al.(2020). b) Ours - Shen et al. (2020). c) Ours - Lösing et al.(2020). d) Ours - Martos et al.(2017), with colors ranging from light yellow to dark red indicating low to high differences, is generated using the Antarctic Ice Shelf and Antarctic Coastline map of Mouginot et al. as a base map (Mouginot et al.,2017).**






On a local scale, our model identifies two substantial regions of heat flow anomalies in the central East Antarctica: the Vostok Subglacial Highlands and the Gamburtsev Subglacial Mountains, both with heat flow values surpassing 80 mW m⁻². The high heat flow in the Vostok Highlands may hint at underlying magmatism at the bottom of the lake, a feature that may explain the significant enhancement of ice-sheet melting and melt-water generation in the region (Artemieva, 2022), and suggests that the

East Antarctica is not exclusively dominated by the cold, stable Craton lithosphere (Shen et al., 2018). In contrast, the heat flow distribution in the West Antarctica displays greater fluctuation. In instance, in the Pine Island Glacier and Byrd Subglacial Basin regions, the GHF anticipated by our model is much lower than that indicated by other models. This result may reflect variances in model sensitivity to local geology characteristics or be related to the underrepresentation of high heat flow samples in the training data.


### 4.5 Uncertainty

To quantify the prediction uncertainties of the model, we adopted the mean standard deviation of ensemble predictions as the metric and further incorporated 446 Antarctic GHF data points compiled by Burton-Johnson et al. (2020) for validation. Although these data, estimated from temperature gradients within ice caps or loose sediments, are subject to high uncertainties

due to interference from climatic forcing, hydrothermal circulation, and ice dynamics (Fisher et al., 2015), they nonetheless provide important support for deepening the model's understanding of the distribution of Antarctic GHF. Based on this, we present quantitative comparisons of prediction uncertainty (see Fig. 8).

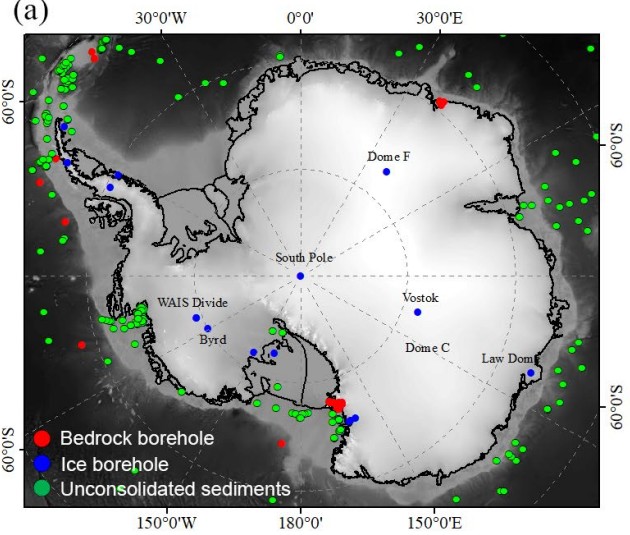

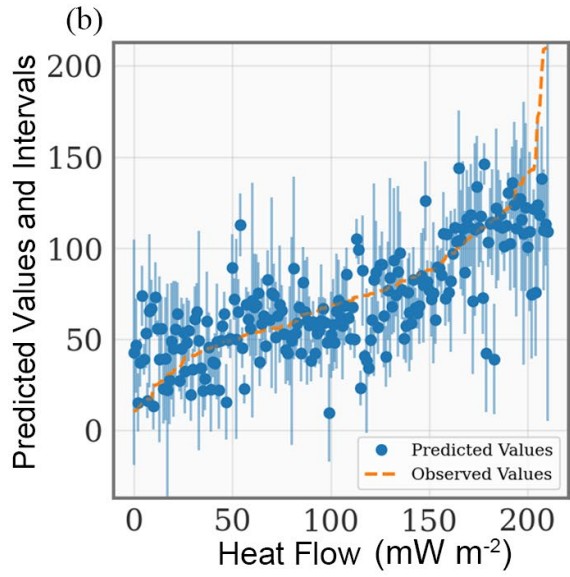





**Figure 8. Location of heat flow in Antarctic continental boreholes and results of uncertainty quantification. (a) Distribution of Antarctic GHF borehole validation sites, categorized as bedrock boreholes (red), ice boreholes (green), and unconsolidated sediment sites (blue). Data source: Burton-Johnson et al. (2020). (b) Ordered prediction interval comparison between predicted values (blue dots) and observed values (orange line), with blue shading representing the standard deviation of predictions. The horizontal axis denotes heat flow values (0 – 200 mW m⁻²), while the vertical axis ranks predicted and observed values, highlighting model prediction**
**biases and uncertainties.**

The results of the research reveal that the model's uncertainty estimates in heat flow prediction are generally reasonable, with the confidence interval encompassing the majority of observed values (as shown in Fig 8b). Specifically, the predicted values align well with observations in low-heat-flow regions (0–100 mW m⁻²), whereas significantly higher uncertainties emerge in
extreme-value regions (heat flow exceeding 150 mW m⁻²). This discrepancy may stem from the uneven distribution of training data particularly the scarcity of ultra-high heat flow samples or could reflect inherent limitations in the model's architecture to fully capture complex geological features.

## 5 Discussion

In this study, we apply a DNN framework to predict Antarctic GHF. To optimize the model performance, we automatically
adjusted the hyperparameters, including the number of hidden layers, the number of neurons per layer, the activation function, the optimization approaches , the batch size, and the learning rate, through the PSO algorithm. The model also achieved high prediction accuracy in data-sparse areas. However, there are still significant inconsistencies in the results of the present model compared to prior investigations. For example, in the Thwaites Basin in the middle West Antarctica, Schroeder et al. (2014) calculated a GHF of 114 ± 10 mW m⁻², while the present analysis anticipated an average value of 87.2 mW m⁻². This
discrepancy may result from the heterogeneity of local geologic features, differences in raw data processing methods, or the influence of complex processes such as shallow water circulation and unsteady convection in the lithosphere, and further studies are needed to elucidate the underlying mechanisms.

A deeper understanding and quantitative assessment of subglacial GHF in Antarctica necessitates refined analysis of crustal geological characteristics and their inherent complexity. Conventional studies, constrained by insufficient observational heat
production data, have often oversimplified or neglected these factors. However, recent research demonstrates that crustal heat generation plays a non-negligible role in the modeling of subglacial heat flow (Li & Aitken, 2024), driving interdisciplinary integration between glaciology (including observations and modeling) and subglacial geology. At the same time, Antarctic bedrock boreholes face great challenges, with measurements now available only in a few ice-free or subglacial regions. These data mainly reflect localized temperature structures and are highly uncertain because most boreholes fail to reach solid bedrock
and estimate heat flow only from temperature gradients within ice caps or loose sediments, which are susceptible to climate forcing, hydrothermal circulation, and ice dynamics (Fisher et al., 2015). Therefore, direct validation of data becomes a substantial bottleneck in current research.



Recent breakthroughs in Interpretable Machine Learning (IML) and Explainable AI (XAI) (Gunning & Aha, 2019; Murdoch et al., 2019) have opened new avenues for deciphering the "black-box" nature of deep learning models. While deep learning

outperforms conventional simplistic models in predictive accuracy, its opaque decision-making process hinders intuitive understanding of feature importance and directional influences (Dramsch, 2020).To bridge this gap, validating the compatibility of model mechanisms with current geologic information can be beneficial in boosting their credibility (Dwivedi et al., 2023), while offering new routes for studying and interpreting complicated linkages in geoscientific data. By understanding the process of machine learning models, we can get insights into how diverse input features interact and

influence geoscientific events, including relationships that may be difficult to discover through conventional analyses (e.g., Ham et al., 2023; Jiang et al., 2024).

Future research should emphasize the collecting of more high-quality field GHF data to validate and refine model prediction results, especially in places with complicated geology conditions in the East Antarctica. In terms of model improvement, one interesting route is to add physical restrictions into the activation function design to make the model outputs more physically

consistent, and the development of this technique is to be expected. In addition, a posteriori interpretation of model outputs in conjunction with interpretability assessments is also crucial. By integrating an interdisciplinary approach of glaciology, geology, and artificial intelligence, this study provides a new technological pathway for accurate estimation of the Antarctic GHF, which is expected to provide critical basic data support for ice sheet dynamics research and global climate change prediction.

**6 Data Availability**

The heat flow database used in this study is sourced from the following repositories: The IHFC Global Heat Flow Database is available at https://ihfc-iugg.org/products/global-heat-flow-database/data (Global Heat Flow Data Assessment Group,2024). The NGHF dataset from Lucazeau (2019) can be obtained here: https://doi.org/10.1029/2019GC008389. The borehole data can be accessed from Burton-Johnson et al. (2020), at https://github.com/RicardaDziadek/Antarctic-GHF-DB.The geophysical

features employed for model training are detailed in Table 1. Visualization results were generated using ArcGIS. Our GHF dataset in this paper is available at https://zenodo.org/records/15254076(Tang et al., 2025). The python code used to generate the maps is available at https://github.com/alibdsd/Antarctica_GHF_PSO_DNN.

**7 Conclusions**

In this study, a DNN model based on particle swarm optimization is developed for predicting Antarctic GHF, and a continent-

wide scale GHF map is generated by combining the global heat flow dataset and multi-source geological features. Through regional density experiments, we found that PSO-DNN is far superior to the commonly used linear regression method in terms of prediction accuracy and nonlinear modeling ability in areas where data is scarce.Subsequently, the model is applied to the



Antarctic continent. The prediction results show that the heat flow values in East Antarctica are generally low (30–60 mW m⁻²), but there are heat flow anomalies (>80 mW m⁻²) in some local areas (such as the Vostok Subglacial Highlands and the

Gamburtsev Subglacial Mountains), which may be related to deep tectonic activities. In West Antarctica, high heat flows (>60 mW m⁻²) are dominant, concentrated in regions such as the Transantarctic Mountains and the coast of the Amundsen Sea, which is consistent with the active geological structures. Compared with previous studies, the results of this model are most similar to the distribution characteristics of the magnetic data model by Martos et al. However, the extreme values of the GHF in East Antarctica are lower, and the predicted values in some areas of West Antarctica(such as the Thwaites Basin) are lower

than the existing estimates. For example, the predicted value in this study is 87.2 mW m⁻², while the estimated value by Schroeder et al. (2014) is 114 ± 10 mW m⁻². These differences may be due to the exclusion of extreme values during the data preprocessing process or the influence of local geological complexity. The uncertainty analysis shows that the 95% confidence interval of the model prediction covers most of the observed value. However, the uncertainty is higher in the areas of extreme values, reflecting the uneven distribution of the training data and the limitations of the model framework.

By integrating geophysical data and artificial intelligence approaches, this study not only verifies the application potential of neural networks in environments with sparse data but also provides new insights into the spatial variability of the GHF in Antarctica. However, the current results are still limited by the scarcity of in-situ data and the lack of model interpretability. Future research should prioritize obtaining more high-quality borehole data to improve prediction accuracy, especially in areas with complex geological conditions in East Antarctica. At the same time, efforts should be made to explore the introduction

of physical constraints and interpretability analysis to enhance the physical consistency and scientific credibility of the model. These improvements will further promote the progress of research on ice sheet dynamics, subglacial hydrology, and global sea-level changes. With the increasing application of neural networks in Earth system science, this study provides a reference for further exploration and optimization of this approach, demonstrating broad application prospects.

**8 Author contributions.**

TX and LS designed the experiments. LS and WL developed the model code and performed the simulations. All authors commented on and edited drafts of this paper.

**9 Competing interests**

The authors declare that they have no conflict of interest.

**10 Acknowledgments.**

This study is supported by the National Natural Science Foundation of China under Grant (42276257).



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
