# Peer review of "Mapping Antarctic Geothermal Heat Flow with Deep Neural Networks optimized by Particle Swarm Optimization Algorithm"

_EGUsphere, 2025_

## Referee Comment (RC2)

**Review of, "Mapping Antarctic Geothermal Heat Flow with Deep Neural Networks optimized by Particle Swarm Optimization Algorithm", by Liu et al.**

Review by Michael Wolovick

**Summary**

In this manuscript, Liu and coauthors infer Antarctic Geothermal Heat Flow (GHF) by training a neural network on a global database of heat flow measurements and a variety of geological or geophysical predictor variables. They train their model to predict spatially binned global observational GHF data using the input variables, without any knowledge of spatial structure (other than the spatial structure implicit in the predictor variables). They use a technique called particle swarm optimization to select the optimal hyperparameters in their deep learning model. They try four different configurations of this technique to ensure that their results are robust, and they also test the robustness of their model by leaving out variable amounts of input data in a densely sampled region of Europe. Their analysis reveals significant advantages of deep learning techniques as compared to linear regression. Their final predicted Antarctic GHF map is broadly reasonable from a geophysical perspective, with higher heat flow in West Antarctica and the Antarctic Peninsula compared with East Antarctica, although they do find some notable local maxima of GHF in East Antarctica under the Gamburtsev Subglacial Mountains and Vostok Subglacial Highlands. They conclude with a comparison of their model against four other published estimates, and against the limited in situ observational data available.

**Major Comments**

My expertise is in glaciology, numerical modeling, and geophysics, so I cannot truly evaluate the deep learning techniques that the authors have used. As far as I can tell, they appear to have been quite thorough from a machine learning perspective, with a lot of effort having been spent to ensure that their model fits the data well with the right set of hyperparameters. However, I do know something about Antarctic heat flow, and I do understand the importance of having good input datasets and predictor variables for a project like this, especially since their deep learning models have no internal knowledge of spatial relationships: their model treats each grid cell as an independent data point, and therefore any spatial structure in the output must come from spatial structure in the inputs. I also understand how vital it is that the input datasets contain realistic and accurate data underneath the Antarctic Ice Sheet; no matter how good the neural network is, if it is supplied with bad inputs, then it will produce bad outputs. It is therefore vital that the authors manuscript shows readers maps of all of the predictor variables, both globally and with a south polar view for Antarctica.

Unfortunately, this manuscript does not show those maps. It is therefore impossible for readers to evaluate how much the authors' results can be trusted. These figures need not all be in the main text; with 16 variables (14 after removal of two highly colinear variables) there is a lot of information to display, so supplemental figures or an appendix would be fine here. However, it is impossible to properly evaluate the authors' model without seeing those inputs.

This is an especially important issue given the fact that at least two of the input variables are problematic in Antarctica. The "rock type" variable (which is one of only 3 that the authors show, in Fig 2), classifies the entire Antarctic Ice Sheet under the category of "ice". This of course makes sense for a dataset which represents the surficial rock type of a region, but we are interested in the heat flow *underneath* the ice. The subglacial rock type is generally unknown for most of Antarctica, so this dataset is worthless. Additionally, sediment thickness is another variable they use that is poorly constrained underneath most of the Antarctic Ice Sheet. While it is true that active-source seismic surveys have constrained sedimentary basin depths in a handful of locations, for the most part we simply have no idea how thick the sediment is underneath the ice, so this

dataset is also worthless. The authors made decisions about which variables to keep and which to exclude on the basis of colinearity (sensibly choosing to discard redundant variables that were highly correlated with other variables), but they seemed not to have considered physical plausibility or under-ice uncertainty in their decision-making. Good input datasets for a project such as this one must be datasets for which the structure in Antarctica is well-constrained. For at least two of the inputs, that is categorically not the case. The authors need to look over their input datasets and remove those that are not well-constrained underneath the Antarctic Ice Sheet. Most obviously this includes rock type and sediment thickness, but they should double-check all of their inputs to ensure that they have realistic spatial structure in Antarctica. This will require that the authors retrain their deep learning models using only the datasets that are reliable in Antarctica. Unfortunately, this may degrade the quality of the fit and reduce predictive capacity in the rest of the world. However, that is simply the nature of the problem we are trying to solve. If the goal is to infer GHF in Antarctica, then there is no point in using datasets that are unconstrained there.

In addition to the requirement that the input datasets be well-constrained in Antarctica, it is also important that they be free of spatial artifacts, since the authors' deep learning model has no internal knowledge of spatial relationships; it treats every grid cell as an independent data point, and thus it relies on the input datasets to produce spatial structure. Unfortunately, the authors' output model (Fig 6) contains pronounced meridional stripes radiating out from the South Pole. This is likely a result of the fact that the authors interpolated all of their inputs onto a latitude-longitude grid with constant grid spacing. Constant grid spacing in lat-lon space works fine in the mid-latitudes, but it can produce artifacts near the poles, and the authors' result clearly has such artifacts. Since the authors' deep learning model treats every grid cell as an independent data point, it follows that these meridional stripe artifacts in the output are a result of similar stripe artifacts in at least one of the inputs.

There are three main methods that they could use to fix this: 1) they could use projected x/y coordinates for their Antarctic prediction while keeping lat-lon coordinates for the rest of the globe, although this potentially introduces problems in applying a deep learning model trained on lat-lon data to a new set of x/y data if the statistical distributions of the two datasets are different; 2) they could use variable grid spacing in longitude, with more grid points in each row near the equator and fewer grid points in each row near the poles, a method that looks especially attractive given that their deep learning models treat the data as a list of independent points rather than a structured grid anyway; 3) they could keep their regular lat-lon grid but apply latitude-dependent smoothing in the longitude dimension in order to ensure that their input datasets have constant spatial resolution even as the grid converges near the poles[1]. The exact method is up to the authors' choice, and they are of course free to choose a different method from the three that I propose here, but it is important that they appropriately pre-process their predictor variables to remove artifacts in polar regions, because their deep learning algorithm is not going to be capable of removing those artifacts on its own. And, of course, it is vital that the authors *show* us these predictor variables, so that we can verify for ourselves that they are indeed artifact-free.

My overall recommendation is that this paper needs major revisions. I chose major revisions rather than minor mostly because I am recommending that the authors retrain their models after removing datasets that are unconstrained in Antarctica and pre-processing to remove meridional artifacts. The manuscript itself might not need a great many changes. The additional figures I requested showing the input datasets can be placed in a supplement or appendix rather than the main text, and most of the main text can probably be kept without too much change. It might very well be that the new model has a broadly similar distribution of GHF, just without the artifacts. However, I want to see the authors' models retrained after the changes to the inputs that I described above, and since I am recommending that the authors redo their main modeling work, I classify this as a major revision.
* * *
1  In other words, they can apply a separate low-pass filter to every longitude row of their 0.5°x0.5° gridded datasets, with a wavelength in degrees equal to 0.5/cos(lat) and periodic boundary conditions in longitude. This would ensure that their grids have the same effective spatial resolution at high latitudes as they do at the equator.

**Minor Comments**

L29-30: "As an important heat source beneath the Antarctic ice sheet, GHF directly affects the hydrological system under the ice sheet (Kang et al., 2022)."
 While I appreciate the reference to a paper I am coauthor on, there is probably a better reference to use here. We didn't talk much about hydrology in that paper, although we did show basal melt rates.

L33-34: "In addition, the complex interaction between GHF and climate results in a significant degree of variation in Antarctic ice mass distribution."
 I'm not sure what exactly you mean here. How does GHF interact with climate? This sentence needs to be reworded or clarified.

L37-38: "...lays a significant factor for understanding the feedback mechanisms produced by Antarctic ice mass loss and predicting sea-level change"
 This sentence also needs to be clarified.

L40-41: "However, the sparse and uneven distribution of in situ borehole data for GHF, coupled with the severe climatic challenges of direct measurements in the Antarctic continental interior, presents significant challenges for data acquisition (Fisher et al., 2015).
 This sentence should be rephrased. How does the sparse distribution of borehole data present a challenge to data acquisition? It would be more accurate to say that the challenges of data acquisition result in a sparse distribution of data. Perhaps rephrase as, "Unfortunately, the severe logistical difficulties involved in collecting direct measurements in the Antarctic continental interior ensure that the distribution of in situ borehole data for GHF is sparse and uneven (Fischer et al., 2015)."

L42-48: " Conventional approaches fall into two categories: one based on the derivation of geothermal processes, such as decreasing west-to-east heat flow derived from some assumptions of geological conditions (Pollard et al., 2005), crustal and upper-mantle heat flow inferred from seismic models (Shapiro & Ritzwoller, 2004; Shen et al., 2020; Hazzard & Richard, 2024), and Curie temperature depths estimated using satellite magnetometry and thermal models (Maule et al., 2005; Martos et al., 2017). The other was from statistical methods such as multivariate similarity analysis (Stål et al., 2021), Bayesian inversion of multiple datasets (Lösing et al., 2020) and machine learning (Lösing & Ebbing, 2021)."
 These sentences need to be reworked as well. It is wrong to describe the first set of sources as "deriv[ing] geothermal processes". "Geothermal processes" is an ambiguous phrase that could be misinterpreted as referring to hydrothermal circulation, which none of these sources represent. In addition, many of the sources in the first category are also engages in some form of statistical modeling, not process modeling. Shaprio and Ritzwoller, for example, use a similarity function to relate seismic structure in Antarctica to seismic structure elsewhere in the world, where GHF observations are available. They don't perform any thermal modeling. It would be better to say that the first group use one type of data (usually seismic tomography or magnetic anomalies), which the second group use multiple types of data. In addition, there are some references missing here.
 Perhaps this section could be rephrased as: "Conventional approaches fall into two categories: on the one hand are those which use a single type of observation to infer GHF, most commonly seismic tomography (Shapiro & Ritzwoller, 2004; An et al., 2015; Lucazeau, 2019; Shen et al., 2020; Haeger et al., 2022; Hazzard & Richard, 2024) or magnetic anomalies (Maule et al., 2005; Purucker et al., 2012; Martos et al., 2017), although broad tectonic reconstructions have been used as well (Pollard et al., 2005). On the other hand, there are a newer set of statistical methods which integrate multiple types of observational constraints to infer GHF using multivariate

similarity analysis (Stål et al., 2021), Bayesian inversion, (Lösing et al., 2020) or machine learning (Lösing & Ebbing, 2021)."

L60: "...deep learning algorithms … due to its high accuracy…"
	Should be "due to their high accuracy".

L91-92: " Subsequently, these filtered, high-quality point measurements were aggregated by calculating the mean value within a $0.5° \times 0.5°$ latitude-longitude grid. "
	See my major comments about the problems with using a regular lat-lon grid when studying the polar regions.

Figure 1
	Would it be good to include a couple sentences talking about the overall geographic distribution of the global data used to constrain the model?  By eye, these data seem to be heavily biased towards wealthy countries, with much lower data density in Africa, South America, and the Middle East.
	In addition, the color scale should be changed.  Blue-white-red is appropriate for data that represent anomalies with respect to a mean or zero value.  The GHF measurements being shown here are all positive, however, so a different color scale should be used.

Table 1
	As I discussed in the major comments, **all** of these geophysical features need to be shown to the reader, both in global view and in south polar view.  These figures can be placed in a supplement or appendix if necessary.
	In addition, some of these data inputs have two sources listed.  What does it mean when two sources are listed?  Does that mean that the dataset is the mean of both sources?  Or is it the case that one source is a publication and the other is a link to the actual dataset?

L115-116: "Sedimentary layers, due to their low thermal conductivity, act as an insulating blanket, significantly influencing the dissipation of deep-seated heat"
	That may be true, but unfortunately, we have no meaningful constraint on sediment thickness underneath the ice sheet, at least not on a large scale.  That is the challenge for a project like this:  useful datasets are not merely those that have a meaningful physical relationship with heat flow, but those that have a meaningful relationship with heat flow **and** which are well-constrained in Antarctica.  Excluding sedimentary thickness will, no doubt, reduce the quality of the global fit.  However, the challenge of a project like this is to generate a model that can explain global heat flow *using only variables that are known and well-constrained in Antarctica*.  Any predictive power added by sedimentary thickness will be of no help in Antarctica.

L123-125: "The Global Lithological Map (GLiM) database (Hartmann & Moosdorf, 2012) provides surface rock type data, explaining spatial variations in thermal conductivity."
	Same concern as above.  Their map (at least as shown in your Fig 2) lists the entire Antarctic Ice Sheet as the "ice" rock type, which is useless for inferring subglacial heat flow.

L127-128: "To ensure dataset consistency, all predictor variables were resampled to a uniform $0.5° \times 0.5°$ grid using Ordinary Kriging."
	As I discussed in my major comment, a uniform lat/lon grid can produce meridional stripe artifacts near the poles.  Potential solutions include: 1) using projected x/y coordinates in Antarcitca; 2) using uneven grid spacing in longitude; 3) using latitude-dependent smoothing in the longitude dimension.  Or perhaps a different solution that I haven't thought of.  But regardless, something has to be done to help this uniform lat-lon grid perform better near the South Pole.

L157: "the Adam optimizer"
    Does this need a reference?

Equation 3
    I thought that R² was the squared correlation coefficient? The formula for that would be:

$$R^2 = \left( \frac{\sum\limits_{i=1}^{n} (y_i - \bar{y})(\hat{y}_i - \bar{\hat{y}})}{\sum\limits_{i=1}^{n} (y_i - \bar{y})^2 \sum\limits_{i=1}^{n} (\hat{y}_i - \bar{\hat{y}})^2} \right)^2$$

Am I wrong about that? Is this a different definition of $R^2$?

Figure 4
    There is not much range on the y-axis here. Does that mean that all four of the configurations tested here have roughly the same performance? Or that the final result is relatively insensitive to the hyperparameters? In any case, the text should probably discuss the narrow range at some point.

Figure 5
    Why does the circle enclosing your test region include parts of the Black ans Aegian Seas? You have excluded marine observations from your dataset, so it seems like you could make a better dense test region by shifting the circle to only cover terrestrial parts of Europe.
    In addition, why is $R^2$ negative for the linear regression model? Is this a function of the fact that you have defined $R^2$ differently than normal?

L318-322: "However, in the Gamburtsev Subglacial Mountains, Vostok Subglacial Highlands, and the area around Subglacial Lake Vostok, there is an increasing trend of heat flow values, which shows that these regions may have been affected by deep tectonic activity or localized heat sources (Artemieva, 2022)."
    My own inversion for GHF (Wolovick et al, 2021) also showed a local maximum of GHF in the Gamburtsev Mountains which is necessary to fit observations of subglacial water networks there.

Figure 6
    The meridional stripe-artifacts are quite prominent in the final result and uncertainty estimate here. In addition, it would be nice if the uncertainty estimate made some attempt to account for the uncertainty in the input datasets, which have uneven spatial resolution in Antarctica even for variables that are relatively well-constrained like seismic velocity or Curie Depth.

Figure 7
    It would be better to show signed difference rather than absolute difference here. It is important to know which estimate is hotter! This would be a good place to use the blue-white-red color scale from figure 1.
    In addition, there are quite a few additional published estimates that you could compare your model against. Additional comparison datasets include: Shapiro and Ritzwoller (2004); Maule et al., (2005); Purucker et al., (2012); An et al., (2015); Lucazeau, (2019), Haeger et al., (2022); Hazzard and Richards, (2024).

L356: "In instance…"
    Should be, "For instance…"

Figure 8
It appears that many of the observations that you use to validate your model are actually located on the seafloor around Antarctica. While it certainly makes sense to include these data points when so few in situ observations are available, does it really make sense to compare your model against these data when you excluded marine observations from your training data?

L389-392: "This discrepancy may result from the heterogeneity of local geologic features, differences in raw data processing methods, or the influence of complex processes such as shallow water circulation and unsteady convection in the lithosphere, and further studies are needed to elucidate the underlying mechanisms."
In addition, the discrepancy between your results and those of Shroeder et al. (2014) could be the result of model assumptions made by Schroeder et al. They made very specific and potentially limiting assumptions about the form of the subglacial hydrological system when constructing their inverse model, and those assumptions could potentially introduce errors into their result.

Section 6 Data Availability
This section should be after the Conclusions section, not before it.

L426: Zenodo link
It would be nice if this link also contained the processed and gridded datasets used as input to your model. While it is true that these datasets are all available at their original sources, it would be nice if it were possible for interested users to access the gridded inputs that you created for your model at one place.

L437: "...which is consistent with the active geological structures."
Rephrase this, this sounds awkward. Perhaps try: "...which is consistent with the locations of present-day tectonic and volcanic activity."

References:
The references should be in alphabetical order, not in citation order.

**Review References**

An, M., Wiens, D. A., Zhao, Y., Feng, M., Nyblade, A., Kanao, M., Li, Y., Maggi, A., and Lévêque, J.-J.: Temperature, lithosphere-asthenosphere boundary, and heat flux beneath the Antarctic Plate inferred from seismic velocities, Journal of Geophysical Research: Solid Earth, 120, 8720–8742, https://doi.org/10.1002/2015JB011917, 2015.
Haeger, C., Petrunin, A. G., and Kaban, M. K.: Geothermal Heat Flow and Thermal Structure of the Antarctic Lithosphere, Geochemistry, Geophysics, Geosystems, 23, e2022GC010501, https://doi.org/10.1029/2022GC010501, 2022.
Lucazeau, F.: Analysis and Mapping of an Updated Terrestrial Heat Flow Data Set, Geochemistry, Geophysics, Geosystems, 20, 4001–4024, https://doi.org/10.1029/2019GC008389, 2019.
Purucker, M. E.: Geothermal heat flux data set based on low resolution observations collected by the CHAMP satellite between 2000 and 2010, and produced from the MF-6 model following the technique described in Fox Maule et al. (2005), 2012.
Wolovick, M. J., Moore, J. C., and Zhao, L.: Joint Inversion for Surface Accumulation Rate and Geothermal Heat Flow From Ice-Penetrating Radar Observations at Dome A, East Antarctica. Part I: Model Description, Data Constraints, and Inversion Results, Journal of Geophysical Research: Earth Surface, 126, e2020JF005937, https://doi.org/10.1029/2020JF005937, 2021.

---

## Author Comment (AC1)

**Reply on RC1.**

**Dear editor and reviewers:**

**We would like to express our sincere gratitude to you for your thoughtful comments and constructive suggestions of our manuscript, which clearly help us improve the manuscript. Please find our replies below. The reviewer's comments are shown in black, and our responses are in red.**

**Kind regards,**

**Authors**

**Reviewers Comments:**
**Reviewer: 1( Stål, Tobias)**
**Comments to the Authors:**

The study demonstrates systematic optimization of neural network architecture using PSO for hyperparameter tuning, making it a significant methodological advancement over some previous work. The systematic optimization represents a robust alternative to ad-hoc tuning methods commonly used in Earth sciences applications, particularly for such challenging problems as geothermal heat flow prediction in data-sparse regions. The research validates the established understanding of Antarctic thermal structure by confirming the East-West pattern, with predominantly low heat flow values (30-60 mW m$^{-2}$) in East Antarctica and higher values (>60 mW m$^{-2}$) in West Antarctica. This consistency with previous studies strengthens confidence in Antarctic crustal thermal architecture. The combination of automated optimization and independent validation of the first-order approximation of the heat flow distribution makes this work valuable for advancing predictive modelling in polar geophysics.

As an output model of actual geothermal heat to expect and include, e.g., interdisciplinary models, I am more skeptical. I have several questions regarding the observables used (listed below). 1.The authors primarily include legacy data (e.g., as available when Aq1 was generated six years ago) along with a few additional datasets that I believe are not very robust. Some choices are not geologically meaningful, as outlined below, and the lack of qualitative assessment of the observables unfortunately invalidates the otherwise sensitive tests conducted. PSO is a valuable tool for DNN, and transparent enough to generate meaningful uncertainty metrics. However, the robustness that PSO is otherwise known for doesn't really help if the features are not meaningful, and we are treating interpolated grid values with the same weight as high-quality and representative observations (discussed by Al-Aghbary et al, 2025, link below). In general, gradient-based optimizers often outperform PSO in similar setups; however, there is certainly a value in testing and comparing various methods, and I believe there will be more development in this field over the coming years, including hybrid strategies (as introduced here with the Adam optimizer).

The ROI analysis offers a reasonable approach to address variations in in-situ data point density; however, a fundamental problem persists regarding how well a single heat flow measurement can represent an entire grid cell. Studies from West Antarctica demonstrate very large local variations in geothermal heat flow. While averaging measurements within global database cells could theoretically mitigate this issue, many cells contain only a single measurement. In these cases, we lack insight into the local conditions that were actually sampled, whether the measurement represents typical regional conditions or a localized anomaly. This spatial representativeness problem becomes particularly acute in Antarctica, where individual point measurements must characterize grid cells spanning millions of square kilometres, potentially introducing significant bias into the training dataset through disproportionate weighting of these sparse observations. However, those issues are not for this paper to resolve, and the methods and analysis are communicated very transparently and clearly. The paper contains many insightful comments regarding concerns and limitations, which are very welcome and still rare.

Some sections of the introduction are challenging to read and don't really make much sense, as if they were written by a language model rather than a scientist. The figures are very good; however, I suggest that Fig.7 be updated (as below).

I am supportive of the publication. However, I am not sure that this is the optimal journal, as the paper's main quality lies in the development of the DNN methods; however, I leave this for the editor to consider.

Thank you for your detailed comments. We greatly appreciate the recognition of our systematic PSO-based optimization approach and the transparent communication of methods and limitations. Your constructive suggestions regarding
observable selection and spatial representativeness has helped us significantly improve the manuscript.

Following your suggestions regarding dataset selection, we have updated our datasets using newly available sources, including the Curie depth data for Antarctica from Martos et al., and removed variables that are not geologically meaningful in the Antarctic context (e.g., sediment thickness, rock type, and distance to hotspots). We have also re-
conducted the sensitivity analysis based on the updated dataset. Furthermore, to address the black-box nature of the DNN model, we incorporated Bayesian probabilistic modeling to quantify its internal uncertainty and employed Particle Swarm Optimization (PSO) to optimize the model parameters.

We thank the reviewer for understanding the challenges associated with the inherent spatial representativeness
limitations of the Antarctic heat flow dataset. To mitigate this issue, we applied a quality-weighted averaging method to the grid cells, assigning weights according to the measurement quality ratings. This approach effectively handles grid cells containing multiple measurements; however, as the reviewer rightly pointed out, it remains limited for cells with only a single measurement—this is an unavoidable constraint under the current data conditions and methodology.

We apologize for the introduction sections that were challenging to read. We acknowledge that large language models were used for language polishing in the original manuscript, which may have affected the clarity and scientific coherence of certain passages. This is mainly because our initial draft was written in Chinese, and we lacked confidence in expressing ourselves in the Introduction when translating it into English. Therefore, we used a large language model to assist us with polishing the Introduction. We have now rewritten the introduction section and have updated the figures
accordingly.

**Main Items to Address Before Publication:**

1. A review of the Introduction, correcting some references, and especially ensuring that the text has a clear and meaningful narrative and that statements are supported by discussion and/or references to studies.

Following your suggestion, we have thoroughly revised the Introduction section and corrected the references. The specific revisions are as follows:

*"Geothermal heat flow (GHF) refers to the heat energy transferred from Earth's interior to the surface via conduction or convection (Pollack et al., 1993). As a critical heat source beneath the Antarctic ice sheet, GHF not only directly affects the subglacial hydrological system and promotes basal melting , but also serves as an important boundary*
*condition for numerical models predicting the Antarctic Ice Sheet(AIS) mass balance and global sea-level change (Obase et al., 2023; Pollard et al., 2005; Seroussi et al., 2017; Wearing et al., 2024; Llubes et al., 2006). Furthermore,*

*characterizing the spatial distribution of GHF over Antarctica is crucial for comprehending the continent's past and present tectonic evolution (Reading et al., 2022).*

*Unfortunately, severe logistical challenges associated with collecting direct measurements in the Antarctic interior have resulted in a sparse and uneven distribution of in situ borehole GHF data (Fisher et al., 2015). Conventional approaches fall into two categories: on the one hand are those which use a single type of observation to infer GHF, most commonly seismic tomography (Shapiro & Ritzwoller, 2004; An et al., 2015a; Lucazeau, 2019; Shen et al., 2020; Haeger et al., 2022; Hazzard & Richard, 2024) or magnetic anomalies (Fox Maule et al., 2005; Purucker et al., 2012; Martos et al., 2017), although broad tectonic reconstructions have been used as well (Pollard et al., 2005). On the other hand, there are a newer set of statistical methods which integrate multiple types of observational constraints to infer GHF using multivariate similarity analysis (Stål et al., 2021), Bayesian inversion (Lösing et al., 2020) or machine learning (Lösing & Ebbing, 2021). While these approaches exhibit consistency at the continental scale, characterized by higher GHF beneath West Antarctica and lower values in East Antarctica, substantial discrepancies persist at regional scales. Methods relying on single observation types are typically constrained by limited data resolution and spatial coverage, as well as by underlying assumptions that may lack universal validity. For instance, seismic tomography-based approaches provide regional-average GHF estimates derived from data with limited sensitivity to upper crustal composition and a coarse lateral resolution of 600–1000 km across Antarctica (Shapiro & Ritzwoller, 2004). As demonstrated by Goutorbe et al. (2011) and Lucazeau (2019), integrating multiple observables yields more robust results than those derived from any single dataset. Specifically, Stål et al. (2021) showed that using 14–19 sets of observables produces a misfit of less than 10 mW m⁻², whereas additional datasets may introduce excessive noise without significantly improving estimates. Consequently, multi-observable approaches necessitate a careful selection of features with adequate Antarctic coverage and strict control over the number of inputs. Uncertainties in the original input data can propagate through the modeling process, and the resulting uncertainties in subglacial GHF estimates can substantially impact ice sheet mass balance simulations. Given that Antarctic ice sheet dynamics remain the largest source of uncertainty in future sea-level rise projections, with estimates for the year 2100 under the RCP8.5 scenario ranging from −7.8 to 30.0 cm in multi-model ensembles (Seroussi et al., 2020) to over 1 m when ice-cliff instability is considered (DeConto and Pollard, 2016), reducing GHF uncertainty is critical for improving the reliability of sea-level change predictions.*

*Recently, deep neural networks (DNNs) have emerged as powerful tools for synthesizing high-dimensional geoscience data, leveraging their formidable nonlinear mapping capabilities. Their efficacy has been proven in improving estimates of Antarctic ice sheet surface melt (Hu et al., 2021), estimating sea ice thickness from satellite radiometry (Herbert et al., 2021), and emulating basal melt rates beneath ice shelves (Burgard et al., 2022). However, current neural network models encounter two primary challenges. First, the performance of DNNs is highly sensitive to numerous hyperparameters; manual or suboptimal tuning often leads to poor generalization or overfitting. Second, as inherently opaque "black-box" models, DNNs seldom provide reliable probabilistic estimates or confidence*

*intervals. This lack of quantifiable uncertainty limits their applicability in downstream earth system modeling where error propagation is a concern.*

*To address these issues, this study proposes a hybrid framework that couples DNNs with Particle Swarm Optimization (PSO) algorithms to refine parameter selection, underpinned by a Bayesian module for robust uncertainty*

*quantification. This integrated approach introduces two key processes aimed at enhancing model generalization and reliability. First, the global search capability of PSO is leveraged to optimize DNN hyperparameters, thereby minimizing the objective function and improving predictive accuracy in data-sparse regions. Second, the integration of a Bayesian module facilitates the decomposition of uncertainty into aleatoric components (stemming from input data noise) and epistemic components (inherent in the model architecture and parameters). In the following sections,*

*we detail the dataset construction and methodology, provide an analysis of discrepancies between the new GHF estimates and prior predictions, and discuss potential uncertainties along with their implications for future investigations.”*

2. A discussion on why the datasets are chosen (e.g., "for backward compatibility to be able to compare the results with studies using simplistic statistical approaches", or what you prefer).

Thanks, this suggestion is very helpful. We have substantially revised Section 2.2 to provide a comprehensive rationale for dataset selection. We have also clarified that legacy datasets were included to enable direct comparison with previous studies. Please refer to Section 2.2 of the revised manuscript for details.

3. Reduced dependency on Antarctic in-situ measurements for validation.

We agree. It is important to treat in-situ inferences carefully, since they are representative of localized temperature structure and are potentially susceptible to contamination by thermal signals caused by frictional heating at the base of the ice sheet, hydrological circulation, and local topography. Given the sparsity of Antarctic GHF estimates derived from in-situ temperature probe observations in boreholes and unconsolidated sediment, we have now treated these data as an independent reference rather than as a validation dataset.

4. As Fig. 7, and in text, confirm what is the most similar model and maybe provide some rudimentary numerical test, e.g., average difference.

We thank the reviewer for this suggestion. We have revised Figure 7 to identify the most similar model and have included numerical comparisons to quantify the model similarities.

Quantitative comparison of our model against existing Antarctic GHF estimates reveals that the model of Martos et al. (2017) shows the highest overall agreement with our predictions, with a mean difference of −2.5 mW/m².

5. Check and confirm the correct references, and organize the bibliography.

We have rechecked all references and reorganized the bibliography accordingly. The reference list is now organized alphabetically by first author surname.

**Detailed Comments:**

L37: Mareschal and Jaupart (2013) is a good overview; however, it's not relevant to reconstruct Antarctic tectonic history. Reading et al (2022, NREE) is probably the most suitable example here.

L43: The text here is not clear; I understand, but it needs some editing.

L47: Citing Lösing, Ebbing et al (2020[should be 2021?]) here also appears a bit out of place. Rather, acknowledge how this study helped us contextualise previous temperature-gradient-based studies. Or is it Lösing and Ebbing (2021)?

L50: The statement that "process-based modelling [depends on] complex mathematical formulation" requires some explanation of what this means, how this is a problem, and why the study at hand addresses this.

L50-55: This section is very hard to follow and doesn't really make any sense. It appears to contradict the previous sentence somewhat.

We thank the reviewer for the detailed comments on the Introduction section. We have carefully considered all suggestions and have substantially rewritten this section to improve clarity.

L37: We have revised the citation and replaced Mareschal and Jaupart (2013) with Reading et al. (2022, NREE).

L47: You are correct. The correct citation is Lösing and Ebbing (2021), which we have now corrected.

L43、L50、L50-55: We have removed this confusing section in the revised manuscript and rewritten this section to clarify the meaning. Please refer to the revised manuscript for details.

L61: I respectfully disagree that deep learning has been particularly successful in polar regions. Whilst there have been a few very useful studies recently (Notably by Prof. Tang), and a lot of method development, the general applications of studies have largely been limited by data availability and lack of consistency and structure. Compared to other regions of the world, DL/ML methods in polar regions have often failed to generate outputs that have been widely accepted to advance our understanding. The statement requires some analysis of why Polar regions have been more successful than elsewhere. In general, I believe that extra caution is required, and uncertainty must be communicated well when dealing with the unknown subglacial geology and in interdisciplinary studies. In the past, we've seen many examples of how research outputs have found interdisciplinary applications that they are not suited for. This is due to change, of course.

We agree with this assessment. While deep learning is a powerful tool, its application in polar regions faces certain limitations due to data scarcity and the "black-box" nature of these models. We have revised the text to provide a more balanced view. Improving model interpretability and addressing data constraints remain key directions for future development in this field.

Section 2.1: One major problem for empirical heat flow models, and related models, is that the training set, or reference set, is not an unbiased representation of Earth's surface. Some settings are highly overrepresented (Stål et al, 2022, Frontiers). How does this impact your results?

We acknowledge that this sampling bias represents a limitation that cannot be fully resolved through methodological improvements alone—it requires fundamental expansion of the observational database in underrepresented regions. Neural networks learn weights from large datasets, and when certain geological settings or regions are overrepresented in the training data, this inevitably influences the model's predictions. This bias is also reflected in our results, where very few predicted values exceed 120 mW/m², likely due to the underrepresentation of high heat flow settings in the global training dataset.

Addressing this limitation will require continued efforts to acquire new measurements in undersampled regions and geological settings.

L87: "marine measurements excluded," but Figure 1 map shows many marine measurements in, e.g., the North Atlantic,

Mediterranean, and East China Sea. Are those included or not?

We have re-excluded data with the "marine" domain attribute from the IHFC database. For the NGHF database, we only retained data with geography codes A, B, C, D, E, F, G, and H (representing continental regions: Africa, North America, South America, Australia, Europe/Greenland, miscellaneous lands, Antarctica, and Asia/Arabia/India, respectively), excluding all oceanic measurements. Figure 1 has been updated to reflect this correction.

[Figure]

Figure 1. Spatial distribution of global GHF measurements used for model training.

Fig. 1: "Dataset obtained from IHFC and NGHF"; however, the text only mentions IHFC. Were the two databases merged? Wouldn't that duplicate most records in NGHF?

Yes, the two databases were merged. After excluding marine data, the IHFC database contains 41,511 measurements and the NGHF database contains 44,698 measurements. There is indeed substantial overlap between these two databases. After merging and removing duplicate records, we obtained a final dataset of 57,654 unique heat flow measurements.

[Figure]

[Figure]

Figure 2. Overlap between IHFC and NGHF.

Table 1 (and general comments on features used):

Why are you using such relatively old datasets? With all respect to the legacy, the results from CRUST1, Shaeffer and Lebedev (2015), and An et al. (2015) are all good studies; however, they are over ten years old, and a lot of data have been collected since then. I notice a significant similarity with the observables used to produce Aq1 back in 2019-20; however, I would have used different datasets today.

We acknowledge that some of the datasets used in this study are relatively dated. As noted earlier, to enable direct comparison with previous studies using statistical and machine learning approaches (e.g., An et al., 2015; Martos et al., 2017), we included several legacy datasets that have been standard inputs in Antarctic GHF modeling. However, following the reviewers' suggestion, we have updated several datasets in the revised manuscript. Specifically, we have incorporated the Curie depth data for Antarctica from Martos et al. (2017) to supplement the GCDM model, and removed features that lack physical significance in the Antarctic context (e.g., sediment thickness, rock type, and distance to hotspots). We acknowledge that further improvements could be achieved by incorporating more recent datasets as they become available, and we consider this an important direction for future work.

Rock type is likely not a very useful observable, as most of Antarctica is classified as ice, and we know that the crustal geology is important but challenging to model (Stål et al., 2024, GRL).

We agree. Rock type is not a meaningful observable in Antarctica since most of the continent is classified as ice (see figure below). We have removed this feature from our model in the revised manuscript.

[Figure]

Figure 3. Global Lithological Map

Some observables, e.g., CTD from Li et al (2017), have very little coverage in Antarctica.

Thanks for the comment. We have now augment the dataset with data from Martos et al. (2017).

What depths are the P wave speed and S wave speed taken from? Can the tomographic model suggest values for the crust, as suggested on L117?

The P-wave and S-wave velocities were extracted at 150 km depth from the Schaeffer and Lebedev (2015) model. This depth
is within the upper mantle rather than the crust. We have revised the text to clarify that these seismic velocities serve as proxies for the thermal state of the upper mantle, rather than the crust, and reflect the lithospheric thermal structure that influences surface heat flow.

*"We selected shear-wave velocity (Vs) and compressional-wave velocity (Vp) data at 150 km depth from the Schaeffer and*
*Lebedev (2015) model, which is constructed from cluster analysis of global surface wave tomography without requiring a*
*priori assumptions about Earth's structure, thus objectively reflecting upper mantle velocity anomalies beneath Antarctica."*

Distance to hotspot cites Anderson (2016); however, this study is not in the reference list, only Anderson (1998), which, to my understanding, doesn't provide the spatial data referred to here. Is it the Complete Hot Spot Table? This list, as far as I know,
has not been peer-reviewed, and I am rather sceptical of it. As above, it should probably be regarded as legacy work, as there was very little to constrain some of those suggestions 25-30 years ago.

You are correct, the data source was the Complete Hot Spot Table. Given the concerns regarding its reliability and lack of peer review, as well as its limited applicability in the Antarctic context, we have removed this feature from our analysis in the
revised manuscript.

Distance to Volcanoes and hotspots is, as I understand, not distance-weighted in any way. Hence, heat flow values and target locations are equally linked, e.g., if the distance to the nearest volcano is 2000 km or 20 km. This is not a useful predictor of geothermal heat. The Adam optimizer compounds this issue by learning to exploit statistical correlations between raw distance
and heat flow in the training data, regardless of physical plausibility. Since Adam operates purely on numerical gradients, I suppose it will adjust network weights to minimize prediction error even when the learned relationships violate fundamental geology. This creates a model that may perform well statistically but also might generate physically meaningless patterns.

Thanks for your suggestion. The reviewer is correct that unweighted distance values are not physically meaningful predictors
of geothermal heat, as they treat locations 20 km and 2000 km from a volcano equally in terms of their relationship to heat flow.
Following this suggestion, we have implemented distance weighting using an exponential decay function to better reflect the physical relationship between distance and geothermal influence. Specifically, the thermal influence weight is calculated as:
$$w = \exp\left(-d/\lambda\right),$$
where $d$ is the distance to the nearest volcano (in km) and $\lambda$ is the decay parameter. We tested multiple decay parameters (500 km, 1000 km, and 2000 km) to capture different scales of thermal influence. This approach ensures that nearby volcanic sources have a stronger influence on predicted heat flow, which is more physically plausible than using raw distance values.

L207: What procedure to avoid overfitting is applied? Here, I would urge the authors to consider alternative and informative
metrics of uncertainty. The recent paper by Al-Aghbary et al (preprint link below) would be a good starting point. What uncertainty and error could/should we actually optimize to reduce?

*Thankyou for recommending the work of Al-Aghbary et al. As we stated in the manuscript:*

*"To control model complexity and mitigate overfitting, we implemented a multi-faceted regularization strategy: (1) L2 regularization (weight decay) was applied to penalize large weight magnitudes and encourage simpler model representations; (2) batch normalization was implemented after each hidden layer to stabilize training dynamics and accelerate convergence; (3) dropout layers were incorporated between hidden layers to randomly deactivate neurons during training, reducing co-adaptation and improving generalization; and (4) an early stopping mechanism was established, terminating training if validation loss failed to decrease for 10 consecutive epochs, with model weights corresponding to the lowest validation loss retained.*

*Our analysis reveals that aleatoric uncertainty—arising from inherent observational variability and unresolved geological heterogeneity—constitutes the dominant fraction of total uncertainty across Antarctica. This finding suggests that while our model architecture has sufficient capacity to capture underlying patterns, irreducible noise in heat flow observations and small-scale geological complexity impose fundamental limits on prediction accuracy.*

*We have reviewed the recent work by Al-Aghbary et al. (2025), which demonstrates a promising approach to address this limitation. By applying unsupervised clustering to partition geophysical observables into homogeneous subsets and training dedicated local expert models within a Mixture-of-Experts (MoE) framework, they achieved substantial reductions in aleatoric uncertainty (up to 29% in synthetic datasets and 8% in real-world GHF data) while maintaining stable epistemic uncertainty. This cluster-specific modeling strategy offers a compelling direction for future improvement of Antarctic GHF predictions, particularly given the pronounced geological heterogeneity between tectonically active West Antarctica and stable East Antarctic cratons. We have added discussion of this approach in the revised manuscript as a promising avenue for future work."*

L281: Including the few in situ measurements in Antarctica is very problematic. 1. Most of them don't reach the bed and represent the paleoclimate and hydrology rather than geothermal heat. 2. They are very sparse, and there are no measurements to average in each grid. 3. Some measurements should be treated with some particular care, as they are either old or associated with large technical challenges when made. 4, and most importantly, they get a disproportional weight as they will be very similar to the surrounding region.

Correct! The Antarctic in-situ measurements indeed present the issues you outlined. We have therefore excluded Antarctic heat flow data from our dataset and use these measurements only as reference points for evaluating model performance.

Figure 6: There appear to be gridding artifacts from the projection of observables, such as lineations pointing toward the South Pole.

Thanks for your comment. We have revised Figure 6 to address these artifacts.

L340 and L438: This statement does not seem to agree with Fig. 7 (?). Instead, it appears that your distribution resembles Lösing et al. (2020) most, which I believe should be Lösing and Ebbing (2021).

We have retrained the model and re-performed the comparisons. The result shows that the model of Martos et al. (2017) generally best matches our predictions, with an average difference of -2.5 mW/m². We have updated the text at L340 and L438

accordingly to ensure consistency with Figure 7.

L353 Are the values you get at Lake Vostok simply the in-situ measurement extrapolated? This measurement is likely to get a very high weight.

We thank the reviewer for raising this concern. The elevated GHF values predicted around Subglacial Lake Vostok are not extrapolations from the in-situ borehole measurement, as Antarctic heat flow data were entirely excluded from the training dataset. Our model was trained exclusively on global continental heat flow data outside Antarctica, and the Antarctic predictions are therefore purely extrapolated based on the relationships learned from other continents.

Figure 7. Show the difference, with the sign, rather than the absolute difference.

OK. We have revised Figure 7 as requested, now presenting the difference with its respective sign instead of the absolute difference.

L361 As explained above, I agree with Fisher, as you cite, and I don't think this is a valid test. The measurements are too sparse and represent the very local conditions. The measurements in the interior don't reach the bedrock. We need some further discussion and evidence to claim that the measurements "nonetheless" provide good support. Two papers to consider here are Talalay et al. (2020, Cryosphere) and Mony et al. (2020, Glaciology).

We thank the reviewer for this clarification and for recommending the papers by Talalay et al. (2020) and Mony et al. (2020). We agree that the in-situ measurements are too sparse to serve as a robust validation dataset and represent only very localized conditions. As Talalay et al. (2020) demonstrate, the basal temperature at the Antarctic Ice Sheet and the temperature gradient in subglacial rocks have been directly measured only a few times. Furthermore, Mony et al. (2020) emphasize that thermal gradients within the ice cannot be used to estimate the solid Earth contribution with any certainty unless the exact basal conditions are known and the borehole reaches sufficient depth.

Accordingly, we have revised our approach: these in-situ data are no longer used for model training or validation, but are instead provided solely as an independent reference for qualitative comparison.

The Conclusion is too long and mainly repetitive. I suggest shortening it and merging some items with the Discussion if required.

Thanks for your advice. We have now shortened and streamlined it to focus clearly on the principal findings:

*"In this study, we present an integrated framework combining PSO-optimized deep neural networks with Bayesian uncertainty quantification for predicting Antarctic GHF. Through regional density experiments, we found that our model is significantly outperforms linear regression in terms of prediction accuracy and nonlinear mapping capacity, particularly in data-constrained environments. The resulting GHF distribution reveals a pronounced East-West dichotomy. Elevated heat flow anomalies are concentrated along the coastal margins of West Antarctica, primarily driven by active lithospheric extension and tectonic activity. Notably, our model predicts elevated GHF values in East Antarctica compared to previous studies, suggesting that the East Antarctic Shield may not be as uniformly cratonic or thermally stable as formerly assumed. Furthermore, uncertainty decomposition reveals that aleatoric components dominate the total predictive variance, highlighting the fundamental limits on predictability imposed by inherent observational noise and unresolved small-scale geological heterogeneity. Future research should prioritize the acquisition of high-quality borehole measurements in data-sparse regions and the integration of physics-informed constraints to enhance model interpretability and geophysical fidelity."*

Bibliography:

The list is not organised and rather chaotic. A few key studies appear to be missing, and even some citations in the text are missing. Please check Lösing et al. (2020) and Lösing and Ebbing (2021); I suspect that the papers have been mixed up a few times in the manuscript. Anderson (2016) is also missing or might have been confused by another study.

We apologize for the oversight and the resulting confusion in the reference list. We have reorganized the reference list into alphabetical order by first author's surname, following standard formatting conventions.

I would recommend that the authors have a look at the following suggestions:

Al-Aghbary et al.

(In review, https://www.authorea.com/doi/full/10.22541/au.175373261.14525669)

Mony et al. (2020, Glaciology).

Reading et al (2022, NREE)

Stål et al. (2022, Frontiers).

Stål et al. (2024, GRL).

Talalay et al. (2020, Cryosphere)

We thank the reviewer for these valuable references. We have carefully reviewed all suggested studies and incorporated them into the revised manuscript. These references have significantly improved the quality of our discussion.

Reference:

Reading, A. M., Stål, T., Halpin, J. A., Lösing, M., Ebbing, J., Shen, W., McCormack, F. S., Siddoway, C. S., and Hasterok, D.: Antarctic geothermal heat flow and its implications for tectonics and ice sheets, Nat. Rev. Earth Environ., 3, 814–831, https://doi.org/10.1038/s43017-022-00348-y, 2022.

Stål, T., Reading, A. M., Halpin, J. A., and Whittaker, J. M.: Properties and biases of the global heat flow compilation, Front.

Earth Sci., 10, 963525, https://doi.org/10.3389/feart.2022.963525, 2022.

Stål, T., Halpin, J. A., Goodge, J. W., and Reading, A. M.: Geology matters for Antarctic geothermal heat, Geophys. Res. Lett., 51, e2024GL110098, https://doi.org/10.1029/2024GL110098, 2024.

Mony, L., Roberts, J. L., and Halpin, J. A.: Inferring geothermal heat flux from an ice-borehole temperature profile at Law Dome, East Antarctica, J. Glaciol., 66, 509–519, https://doi.org/10.1017/jog.2020.27, 2020.

Talalay, P., Li, Y., Augustin, L., Clow, G. D., Hong, J., Lefebvre, E., Markov, A., Motoyama, H., and Ritz, C.: Geothermal heat flux from measured temperature profiles in deep ice boreholes in Antarctica, The Cryosphere, 14, 4021–4037, https://doi.org/10.5194/tc-14-4021-2020, 2020.

Al-Aghbary, M., Awaleh, M. O., Jalludin, M., et al.: Improving Geothermal Heat Flow Predictions and Uncertainty Quantification using Clustering-based Quantile Regression Forests, Authorea [preprint], https://doi.org/10.22541/au.175373261.1452

5669/v1, 28 July 2025.

---

## Author Comment (AC2)

**Reply on RC2.**

**Dear editor and reviewers:**

**We would like to express our sincere gratitude to you for your thoughtful comments and constructive suggestions of our manuscript, which clearly help us improve the manuscript. Please find our replies below. The reviewer's comments are shown in black, and our responses are in red.**

**Kind regards,**

**Authors**

**Review of, "Mapping Antarctic Geothermal Heat Flow with Deep Neural Networks optimized by Particle Swarm Optimization Algorithm", by Liu et al.**

**Review by Michael Wolovick**

**Summary**

In this manuscript, Liu and coauthors infer Antarctic Geothermal Heat Flow (GHF) by training a neural network on a global database of heat flow measurements and a variety of geological or geophysical predictor variables. They train their model to predict spatially binned global observational GHF data using the input variables, without any knowledge of spatial structure (other than the spatial structure implicit in the predictor variables). They use a technique called particle swarm optimization to select the optimal hyperparameters in their deep learning model. They try four different configurations of this technique to ensure that their results are robust, and they also test the robustness of their model by leaving out variable amounts of input data in a densely sampled region of Europe. Their analysis reveals significant advantages of deep learning techniques as compared to linear regression. Their final predicted Antarctic GHF map is broadly reasonable from a geophysical perspective, with higher heat flow in West Antarctica and the Antarctic Peninsula compared with East Antarctica, although they do find some notable local maxima of GHF in East Antarctica under the Gamburtsev Subglacial Mountains and Vostok Subglacial Highlands. They conclude with a comparison of their model against four other published estimates, and against the limited in situ observational data available.

**Major Comments**

My expertise is in glaciology, numerical modeling, and geophysics, so I cannot truly evaluate the deep learning techniques that the authors have used. As far as I can tell, they appear to have been quite thorough from a machine learning perspective, with a lot of effort having been spent to ensure that their model fits the data well with the right set of hyperparameters. However, I do know something about Antarctic heat flow, and I do understand the importance of having good input datasets and predictor variables for a project like this, especially since their deep learning models have no internal knowledge of spatial relationships: their model treats each grid cell as an independent data point, and therefore any spatial structure in the output must come from spatial structure in the inputs. I also understand how vital it is that the input datasets contain realistic and accurate data underneath the Antarctic Ice Sheet; no matter how good the neural network is, if it is supplied with bad inputs, then it will produce bad outputs. It is therefore vital that the authors
manuscript shows readers maps of all of the predictor variables, both globally and with a south polar view for Antarctica.

Unfortunately, this manuscript does not show those maps. It is therefore impossible for readers to evaluate how much the authors' results can be trusted. These figures need not all be in the main text; with 16 variables (14 after removal of two highly colinear variables) there is a lot of information to display, so supplemental figures or an appendix would be fine here. However, it is impossible to properly evaluate the authors' model without seeing those inputs.

This is an especially important issue given the fact that at least two of the input variables are problematic in Antarctica. The "rock type" variable (which is one of only 3 that the authors show, in Fig 2), classifies the entire Antarctic Ice Sheet under the category of "ice". This of course makes sense for a dataset which represents the surficial rock type of a region, but we are interested in the heat flow underneath the ice. The subglacial rock type is generally unknown for most of Antarctica, so this dataset is worthless. Additionally, sediment thickness is another variable they use that is poorly constrained underneath most of the Antarctic Ice Sheet. While it is true that active-source seismic surveys have constrained sedimentary basin depths in a handful of locations, for the most part we simply have no idea how thick the sediment is underneath the ice, so this dataset is also worthless. The authors made decisions about which variables to keep and which to exclude on the basis of colinearity (sensibly choosing to discard redundant variables that were highly correlated with other variables), but they seemed not to have considered physical plausibility or under-ice uncertainty in their decision-making. Good input datasets for a project such as this one must be datasets for which the structure in Antarctica is well-constrained. For at least two of the inputs, that is categorically not the case. The authors need to look over their input datasets and remove those that are not well-constrained underneath the Antarctic Ice Sheet. Most obviously this includes rock type and sediment thickness, but they should double-check all of their inputs to ensure that they have realistic spatial structure in Antarctica. This will require that the authors retrain their deep learning models using only the datasets that are reliable in Antarctica. Unfortunately, this may degrade the quality of the fit and reduce predictive capacity in the rest of the world. However, that is simply the nature of the problem we are trying to solve. If the goal is to infer GHF in Antarctica, then there is no point in using datasets that are unconstrained there.

In addition to the requirement that the input datasets be well-constrained in Antarctica, it is also important that they be free of spatial artifacts, since the authors' deep learning model has no internal knowledge of spatial relationships; it treats every grid cell as an independent data point, and thus it relies on the input datasets to produce spatial structure. Unfortunately, the authors' output model (Fig 6) contains pronounced meridional stripes radiating out from the South Pole. This is likely a result of the fact that the authors interpolated all of their inputs onto a latitude-longitude grid with constant grid spacing. Constant grid spacing in lat-lon space works fine in the mid latitudes, but it can produce artifacts near the poles, and the authors' result clearly has such artifacts. Since the authors' deep learning model treats every grid cell as an independent data point, it follows that these meridional stripe artifacts in the output are a result of similar stripe artifacts in at least one of the inputs.

There are three main methods that they could use to fix this: 1) they could use projected x/y coordinates for their Antarctic prediction while keeping lat-lon coordinates for the rest of the globe, although this potentially introduces problems in applying a deep learning model trained on lat-lon data to a new set of x/y data if the statistical distributions of the two datasets are different; 2) they could use variable grid spacing in longitude, with more grid points in each row near the equator and fewer grid points in each row near the poles, a method that looks especially attractive given that their deep learning models treat the data as a list of independent points rather than a structured grid anyway; 3) they could keep their regular lat-lon grid but apply latitude-dependent smoothing in the longitude dimension in order to ensure that their input datasets have constant spatial resolution even as the grid converges near the poles1. The exact method is up to the authors' choice, and they are of course free to choose a different method from the three that I propose here, but it is important that they appropriately pre-process their predictor variables to remove artifacts in polar regions, because their deep learning algorithm is not going to be capable of removing those artifacts on its own. And, of course, it is vital that the authors show us these predictor variables, so that we can verify for ourselves that they are indeed artifact-free.

My overall recommendation is that this paper needs major revisions. I chose major revisions rather than minor mostly because I am recommending that the authors retrain their models after removing datasets that are unconstrained in Antarctica and pre-processing to remove meridional artifacts. The manuscript itself might not need a great many changes. The additional figures I requested showing the input datasets can be placed in a supplement or appendix rather than the main text, and most of the main text can probably be kept without too much change. It might very well be that the new model has a broadly similar distribution of GHF, just without the artifacts. However, I want to see the authors' models retrained after the changes to the inputs that I described above, and since I am recommending that the authors redo their main modeling work, I classify this as a major revision.

We sincerely thank the reviewer for the thorough and insightful evaluation of our manuscript. The reviewer's expertise in glaciology, numerical modeling, and geophysics has been invaluable in identifying critical issues that significantly improve the scientific rigor of this study. We have carefully addressed all major concerns as follows:

1. Display of Predictor Variable Maps. We fully agree that displaying all predictor variables is essential for readers to evaluate the reliability of our results. We have added Figure S1 in Supplementary Materials, which includes global maps of all retained predictor variables and their corresponding Antarctic polar stereographic views (EPSG:3031).

2. Removal of Unreliable Input Datasets. We agree with the reviewer's concern regarding the physical plausibility and uncertainty. We have removed rock type, sediment thickness and distance to hotpot from our predictor variables, as these datasets are indeed poorly constrained beneath the Antarctic Ice Sheet. The model has been retrained using only the reliable datasets. As anticipated, global predictive performance slightly decreased, but this trade-off is necessary and appropriate to ensure reliable Antarctic predictions.

3. Meridional Stripe Artifacts. We thank the reviewer for identifying this issue and for providing detailed suggestions for resolution. Upon investigation, we found that the meridional stripe artifacts were caused by an error in the projected coordinate system during visualization in ArcGIS. The original approach of directly interpolating in the EPSG:4326 coordinate system resulted in artificial patterns near the poles due to grid convergence. We have now corrected this by projecting the data from EPSG:4326 to EPSG:3031 (Antarctic Polar Stereographic Projection) before interpolation. The revised Figure 6 no longer exhibits these artifacts.

**Minor Comments**

L29-30: "As an important heat source beneath the Antarctic ice sheet, GHF directly affects the hydrological system under the ice sheet (Kang et al., 2022)."

While I appreciate the reference to a paper I am coauthor on, there is probably a better reference to use here. We didn't talk much about hydrology in that paper, although we did show basal melt rates.

Thanks for your advice. We have replaced the reference with a more appropriate citations(Siegert et al., 2016):

Siegert, M. J., Ross, N., Li, J., Schroeder, D. M., Rippin, D., Ashmore, D., Bingham, R. G., and Le Brocq, A. M.: Antarctic subglacial groundwater: a concept paper on its measurement and potential influence on ice flow, Geol. Soc. London Spec. Publ., 461, 197–213, https://doi.org/10.1144/SP461.6, 2018.

L33-34: "In addition, the complex interaction between GHF and climate results in a significant degree of variation in Antarctic ice mass distribution."

I'm not sure what exactly you mean here. How does GHF interact with climate? This sentence needs to be reworded or clarified.

L37-38: "...lays a significant factor for understanding the feedback mechanisms produced by Antarctic ice mass loss and predicting sea-level change"

This sentence also needs to be clarified。

L40-41: "However, the sparse and uneven distribution of in situ borehole data for GHF, coupled with the severe climatic challenges of direct measurements in the Antarctic continental interior, presents significant challenges for data acquisition (Fisher et al., 2015).

    This sentence should be rephrased. How does the sparse distribution of borehole data present a challenge to data acquisition? It would be more accurate to say that the challenges of data acquisition result in a sparse distribution of data. Perhaps rephrase as, "Unfortunately, the severe logistical difficulties involved in collecting direct measurements in the Antarctic continental interior ensure that the distribution of in situ borehole data for GHF is sparse and uneven (Fischer et al., 2015)."

L42-48: " Conventional approaches fall into two categories: one based on the derivation of geothermal processes, such as decreasing west-to-east heat flow derived from some assumptions of geological conditions (Pollard et al., 2005), crustal and upper-mantle heat flow inferred from seismic models (Shapiro & Ritzwoller, 2004; Shen et al., 2020; Hazzard & Richard, 2024), and Curie temperature depths estimated using satellite magnetometry and thermal models (Maule et al., 2005; Martos et al., 2017). The other was from statistical methods such as multivariate similarity analysis (Stål et al., 2021), Bayesian inversion of multiple datasets (Lösing et al., 2020) and machine learning (Lösing & Ebbing, 2021)."

    These sentences need to be reworked as well. It is wrong to describe the first set of sources as "deriv[ing] geothermal processes". "Geothermal processes" is an ambiguous phrase that could be misinterpreted as referring to hydrothermal circulation, which none of these sources represent. In addition, many of the sources in the first category are also engages in some form of statistical modeling, not process modeling. Shaprio and Ritzwoller, for example, use a similarity function to relate seismic structure in Antarctica to seismic structure elsewhere in the world, where GHF observations are available. They don't perform any thermal modeling. It would be better to say that the first group use one type of data (usually seismic tomography or magnetic anomalies), which the second group use multiple types of data. In addition, there are some references missing here.

    Perhaps this section could be rephrased as: "Conventional approaches fall into two categories: on the one hand are those which use a single type of observation to infer GHF, most commonly seismic tomography (Shapiro & Ritzwoller, 2004; An et al., 2015; Lucazeau, 2019; Shen et al., 2020; Haeger et al., 2022; Hazzard & Richard, 2024) or magnetic anomalies (Maule et al., 2005; Purucker et al., 2012; Martos et al., 2017), although broad tectonic reconstructions have been used as well (Pollard et al., 2005). On the other hand, there are a newer set of statistical methods which integrate multiple types of observational constraints to infer GHF using multivariate similarity analysis (Stål et al., 2021), Bayesian inversion, (Lösing et al., 2020) or machine learning (Lösing & Ebbing, 2021)."

L60: "...deep learning algorithms … due to its high accuracy…"
    Should be "due to their high accuracy".

(L29-60): We thank the reviewer for the detailed and constructive comments on the Introduction section. Based on the suggestions from both reviewers, we have substantially rewritten this section to address all identified issues:

*"Geothermal heat flow (GHF) refers to the heat energy transferred from Earth's interior to the surface via conduction or convection (Pollack et al., 2013). As a critical heat source beneath the Antarctic ice sheet, GHF not only directly affects the subglacial hydrological system and promotes basal melting , but also serves as an important boundary condition for numerical models predicting the Antarctic Ice Sheet(AIS) mass balance and global sea-level change (Obase et al., 2023; Pollard et al., 2005; Wearing et al., 2024; Llubes et al., 2006). Furthermore, characterizing the spatial distribution of GHF over Antarctica is crucial for comprehending the continent's past and present tectonic evolution (Artemieva, 2011; Reading et al., 2022).*

*Unfortunately, severe logistical challenges associated with collecting direct measurements in the Antarctic interior have resulted in a sparse and uneven distribution of in situ borehole GHF data (Fisher et al., 2015). Conventional approaches fall*

*into two categories: on the one hand are those which use a single type of observation to infer GHF, most commonly seismic tomography (Shapiro & Ritzwoller, 2004; An et al., 2015; Lucazeau, 2019; Shen et al., 2020; Haeger et al., 2022; Hazzard & Richard, 2024) or magnetic anomalies (Maule et al., 2005; Purucker et al., 2012; Martos et al., 2017), although broad tectonic reconstructions have been used as well (Pollard et al., 2005). On the other hand, there are a newer set of statistical methods which integrate multiple types of observational constraints to infer GHF using multivariate similarity analysis (Stål et al., 2021), Bayesian inversion, (Lösing et al., 2020) or machine learning (Lösing & Ebbing, 2021). While these approaches exhibit*

*consistency at the continental scales—characterized by higher GHF beneath West Antarctica and lower values in East Antarctica—substantial discrepancies persist at regional scales. Methods relying on single observation types are typically constrained by limited data resolution and spatial coverage, as well as by underlying assumptions that may lack universal validity. For instance, seismic tomography-based approaches provide regional-average GHF estimates derived from data with limited sensitivity to upper crustal composition and a coarse lateral resolution of 600–1000 km across Antarctica*

*(Shapiro & Ritzwoller, 2004; O'Donnell et al., 2019). As demonstrated by Goutorbe et al.(2011) and Lucazeau (2019), integrating multiple observables yields more robust results than those derived from any single dataset. Specifically, Stål et al. (2021) showed that using 14–19 sets of observables produces a misfit of less than 10 mW m$^{-2}$, whereas additional datasets may introduce excessive noise without significantly improving estimates. Consequently, multi-observable approaches necessitate a careful selection of features with adequate Antarctic coverage and strict control over the number of inputs.*

*Uncertainties in the original input data can propagate through the modeling process, and the resulting uncertainties in subglacial GHF estimates can substantially impact ice sheet mass balance simulations. Given that Antarctic ice sheet dynamics remain the largest source of uncertainty in future sea-level rise projections—with estimates for the year 2100 ranging from −5 to 43 cm of sea level equivalent under high emission scenarios (Seroussi et al., 2020; IPCC, 2021)—reducing GHF uncertainty is critical for improving the reliability of sea-level change predictions.*

*Recently, deep neural networks (DNNs) have emerged as powerful tools for synthesizing high-dimensional geoscience data, leveraging their formidable nonlinear mapping capabilities. Their efficacy has been proven in improving estimates of Antarctic ice sheet surface melt (Hu et al., 2021), predicting seasonal sea ice extent (Andersson et al., 2021), and simulating basal melt rates beneath ice shelves (Burgard et al., 2022). However, current neural network models encounter two primary challenges.*

*First, the performance of DNNs is highly sensitive to numerous hyperparameters; manual or suboptimal tuning often leads to poor generalization or overfitting. Second, as inherently opaque "black-box" models, DNNs seldom provide reliable probabilistic estimates or confidence intervals. This lack of quantifiable uncertainty limits their applicability in downstream earth system modeling where error propagation is a concern.*

*To address these issues, this study proposes a hybrid framework that couples DNNs with Particle Swarm Optimization (PSO) algorithms to refine parameter selection, underpinned by a Bayesian module for robust uncertainty quantification. This integrated approach introduces two key processes aimed at enhancing model generalization and reliability. First, the global search capability of PSO is leveraged to optimize DNN hyperparameters, thereby minimizing the objective function and improving predictive accuracy in data-sparse regions.. Second, the integration of a Bayesian module facilitates the*

*decomposition of uncertainty into aleatoric components (stemming from input data noise) and epistemic components (inherent in the model architecture and parameters). In the following sections, we detail the dataset construction and methodology, provide an analysis of discrepancies between the new GHF estimates and prior predictions, and discuss potential uncertainties along with their implications for future investigations."*

L91-92: " Subsequently, these filtered, high-quality point measurements were aggregated by calculating the mean value within a 0.5° × 0.5° latitude-longitude grid. "
    See my major comments about the problems with using a regular lat-lon grid when studying the polar regions.

We thank the reviewer for raising this concern, which relates to the major comment about potential artifacts from using a regular lat-lon grid in polar regions. As described in our response to the major comments, we have addressed this issue by projecting the data from EPSG:4326 to EPSG:3031 (Antarctic Polar Stereographic Projection) before interpolation.

Additionally, we have visualized all input variables in Supplementary Figure S1, which confirms that the input datasets are free of meridional stripe artifacts in Antarctica.

Figure 1

Would it be good to include a couple sentences talking about the overall geographic distribution of the global data used to constrain the model? By eye, these data seem to be heavily biased towards wealthy countries, with much lower data density in Africa, South America, and the Middle East.

In addition, the color scale should be changed. Blue-white-red is appropriate for data that represent anomalies with respect to a mean or zero value. The GHF measurements being shown here are all positive, however, so a different color scale should be used.

We thank the reviewer for the suggestions regarding Figure 1. We have added a brief discussion of the geographic data
distribution in the text:

"The filtered dataset exhibits significant spatial clustering (Fig. 1). Data coverage is substantially denser in North America, Europe, and parts of East Asia, corresponding to regions with longer histories of geothermal exploration. In contrast, Africa, South America, the Middle East, and Antarctica have markedly sparser coverage, with large areas containing few or no
measurements. This geographic bias poses a challenge for empirical GHF modeling, as the training data are not an unbiased representation of Earth's geological diversity, and certain tectonic settings are overrepresented relative to others (Stål et al., 2022)."

Additionally, we have revised Figure 1 to use a sequential color scale that better represents the continuous positive range of GHF measurements.

[Figure]

Figure 1. Spatial distribution of global GHF measurements used for model training.

Table 1

As I discussed in the major comments, all of these geophysical features need to be shown to the reader, both in global view and in south polar view. These figures can be placed in a supplement or appendix if necessary.

In addition, some of these data inputs have two sources listed. What does it mean when two sources are listed? Does that mean that the dataset is the mean of both sources? Or is it the case that one source is a publication and the other is a link to the actual dataset?

We thank the reviewer for these comments. As described in our response to Major Comment 1, we have added Figure S1 in the Supplementary Materials showing all retained predictor variables in both global view and Antarctic polar stereographic view (EPSG:3031). Some features list two data sources: the first reference refers to data from around the world, while the second provides Antarctic-specific data as a regional supplement. We have clarified this in the revised table:

*" Note: Where two references are listed, the first provides global coverage and the second supplements with Antarctic-specific data offering higher regional resolution."*

L115-116: "Sedimentary layers, due to their low thermal conductivity, act as an insulating blanket, significantly influencing the dissipation of deep-seated heat"

That may be true, but unfortunately, we have no meaningful constraint on sediment thickness underneath the ice sheet, at least not on a large scale. That is the challenge for a project like this: useful datasets are not merely those that have a meaningful physical relationship with heat flow, but those that have a meaningful relationship with heat flow and which are well-constrained in Antarctica. Excluding sedimentary thickness will, no doubt, reduce the quality of the global fit. However, the challenge of a project like this is to generate a model that can explain global heat flow using only variables that are known and well-constrained in Antarctica. Any predictive power added by sedimentary thickness will be of no help in Antarctica.

L123-125: "The Global Lithological Map (GLiM) database (Hartmann & Moosdorf, 2012) provides surface rock type data, explaining spatial variations in thermal conductivity."

Same concern as above. Their map (at least as shown in your Fig 2) lists the entire Antarctic Ice Sheet as the "ice" rock type, which is useless for inferring subglacial heat flow.

L115-116 and L123-125: We thank the reviewer for these important comments. We agree that the datasets for this project must not only have a meaningful physical relationship with heat flow, but also be well-constrained in Antarctica. As shown in Figure 2 (Rock Type), the entire Antarctic continent is classified as "Ice and Glaciers (IG)" in the Global Lithological Map database, which provides no information about the subglacial geology. Similarly, sediment thickness is poorly constrained beneath the Antarctic Ice Sheet on a continental scale. We have removed both rock type and sediment thickness from our predictor variables and retrained the model using only datasets that are well-constrained in Antarctica. As anticipated, this reduced the quality of the global fit slightly, but this trade-off is necessary and appropriate since any predictive power added by these variables would be of no help in Antarctica where they are unconstrained.

[Figure]

Figure 2. Global Lithological Map

L127-128: "To ensure dataset consistency, all predictor variables were resampled to a uniform 0.5° × 0.5° grid using Ordinary Kriging."

As I discussed in my major comment, a uniform lat/lon grid can produce meridional stripe artifacts near the poles. Potential solutions include: 1) using projected x/y coordinates in Antarcitca; 2) using uneven grid spacing in longitude; 3) using latitude-dependent smoothing in the longitude dimension. Or perhaps a different solution that I haven't thought of. But regardless, something has to be done to help this uniform lat-lon grid perform better near the South Pole.

We thank the reviewer for emphasizing this issue. As explained in our response to the Major Comments, we have addressed this problem by adopting the first solution suggested by the reviewer: using projected x/y coordinates for Antarctica. Specifically, we project the data from EPSG:4326 to EPSG:3031 (Antarctic Polar Stereographic Projection) before interpolation. The revised Figure 3 confirms that the Antarctic GHF predictions are now free of these artifacts.

[Figure]

**Figure 3. Predicted GHF distribution and associated uncertainty across Antarctica.**

L157: "the Adam optimizer"
    Does this need a reference?

We agree. We have added the appropriate reference for the Adam optimizer (Kingma & Ba, 2015).

Equation 3 I thought that R2 was the squared correlation coefficient? The formula for that would be:

$$R^2 = \left( \frac{\sum_{i=1}^{n} (y_i - \bar{y})(\hat{y}_i - \bar{\hat{y}})}{\sum_{i=1}^{n} (y_i - \bar{y})^2 \sum_{i=1}^{n} (\hat{y}_i - \bar{\hat{y}})^2} \right)^2$$

Am I wrong about that? Is this a different definition of R2?

Indeed, there are two common definitions of $R^2$ in the literature:

(1) Squared Pearson Correlation Coefficient (as the reviewer described): $R^2 = r^2 = (\frac{\sum_{i=1}^{n} (y_i - \bar{y})(\hat{y}_i - \bar{\hat{y}})}{\sqrt{\sum_{i=1}^{n} (y_i - \bar{y})^2 \sum_{i=1}^{n} (\hat{y}_i - \bar{\hat{y}})^2}})^2$

(2) Coefficient of Determination (used in our manuscript):: $R^2 = 1 - \frac{\sum_{i=1}^{n} (y_i - \hat{y}_i)^2}{\sum_{i=1}^{n} (y_i - \bar{y})^2} = 1 - \frac{SS_{res}}{SS_{tot}}$

In our study, we adopted the second definition (coefficient of determination), which is the standard metric used in regression analysis and machine learning applications for evaluating model performance. Unlike the squared correlation coefficient, the coefficient of determination can yield negative values when model predictions perform worse than simply using the mean of the observed values as a predictor. A negative $R^2$ thus indicates that the model fails to capture the underlying patterns in the data.

We have revised the manuscript to explicitly clarify which definition of R² we employed and added an explanation that R² can become negative when the model predictions are worse than the mean baseline. Please refer to the revised Section 3.5 for details.

Figure 4

There is not much range on the y-axis here. Does that mean that all four of the configurations tested here have roughly the same performance? Or that the final result is relatively insensitive to the hyperparameters? In any case, the text should probably discuss the narrow range at some point.

You are right. We have added this discussion to the revised manuscript:

[Figure]

*"Figure 4 illustrates RMSE trends across 100 iterations for the four configurations. Config2 ($c_1 = c_2 = 2.0$, $m = 20$)*
*achieved the lowest final RMSE of 19.65 mW m⁻², followed by Config1 (19.74 mW m⁻²), Config4 (19.77 mW m⁻²), and*
*Config3 (19.94 mW m⁻²). The symmetric acceleration coefficients setting ($c_1 = c_2 = 2.0$) consistently outperformed the*
*asymmetric configuration. Notably, the narrow performance differences (less than 0.3 mW m⁻² between best and worst*
*configurations) suggest that for this specific GHF prediction problem, the optimization landscape is relatively smooth,*

*allowing all configurations to converge to similar solutions. Nevertheless, Config2 was selected as the optimal*
*configuration for subsequent model training based on its lowest validation RMSE."*

Figure 5

Why does the circle enclosing your test region include parts of the Black ans Aegian Seas? You have excluded marine
observations from your dataset, so it seems like you could make a better dense test region by shifting the circle to only
cover terrestrial parts of Europe.

In addition, why is R2 negative for the linear regression model? Is this a function of the fact that you have defined
R2 differently than normal。

We have re-excluded data with the "marine" domain attribute from the IHFC database. For the NGHF database, we only
retained data with geography codes A, B, C, D, E, F, G, and H (representing continental regions: Africa, North America,
South America, Australia, Europe/Greenland, miscellaneous lands, Antarctica, and Asia/Arabia/India, respectively),
excluding all oceanic measurements. However, some data points near coastlines may still appear because the original
database classification includes "continental (lake, river, etc.)". Figure 1 has been updated to reflect this correction.
Regarding the test region: We have algorithmically re-adjusted the position of the test region circle to ensure it is more
concentrated on areas with the highest density of terrestrial observations, minimizing overlap with marine regions.
And the negative $R^2$ value for the linear regression model is a direct consequence of using the coefficient of determination
definition (as discussed in our response to the reviewer's comment on Equation 3).$R^2$ can become negative when the
model's predictions perform worse than simply using the mean of the observed values as a constant predictor.

[Figure]

Figure 5. Performance of DNN and linear regression methods in experiments with different densities of ROI regions.

L318-322: "However, in the Gamburtsev Subglacial Mountains, Vostok Subglacial Highlands, and the area around Subglacial Lake Vostok, there is an increasing trend of heat flow values, which shows that these regions may have been affected by deep tectonic activity or localized heat sources (Artemieva, 2022)."

My own inversion for GHF (Wolovick et al, 2021) also showed a local maximum of GHF in the Gamburtsev Mountains which is necessary to fit observations of subglacial water networks there.

We thank the reviewer for this comment and for sharing the relevant findings from Wolovick et al. (2021). Following the reviewers' recommendations to remove predictor variables that are poorly constrained in Antarctica, we have retrained our model with the refined variable set. Unfortunately, the updated predictions show elevated GHF values in the area around Subglacial Lake Vostok, but the local maximum in the Gamburtsev Subglacial Mountains identified in the original submission is no longer apparent in the revised model. This difference may reflect the trade-off between global predictive power and Antarctic reliability: by excluding variables that are unconstrained beneath the ice sheet, we may have reduced the model's ability to resolve certain local features.

Figure 6

The meridional stripe-artifacts are quite prominent in the final result and uncertainty estimate here. In addition, it would be nice if the uncertainty estimate made some attempt to account for the uncertainty in the input datasets, which have uneven spatial resolution in Antarctica even for variables that are relatively well-constrained like seismic velocity or Curie Depth.

We thank the reviewer for these comments. As described in our response to the Major Comments, we have addressed this issue by projecting the data from EPSG:4326 to EPSG:3031 (Antarctic Polar Stereographic Projection) before interpolation. Regarding uncertainty estimation, we acknowledge that the uncertainty in input datasets, particularly their uneven spatial resolution in Antarctica even for relatively well-constrained variables like seismic velocity or Curie depth, contributes to the overall aleatoric uncertainty captured in our framework. And we have substantially improved our uncertainty quantification by implementing a Bayesian framework that decomposes total predictive uncertainty into aleatoric and epistemic components:

*"The distribution of uncertainty components (Fig. 8a) reveals that aleatoric uncertainty constitutes the dominant fraction of total uncertainty throughout the study region. The median aleatoric uncertainty ($\sim 650$ mW$^2$ m$^{-4}$) substantially exceeds the median epistemic uncertainty ($\sim 70$ mW$^2$ m$^{-4}$), indicating that inherent variability in heat flow observations and unresolved local geological heterogeneity—including the uneven spatial resolution of input datasets in Antarctica— represent the primary sources of predictive uncertainty. This finding suggests that while our model has sufficient capacity*

*and training data to capture the underlying patterns, the irreducible observational noise and small-scale geological complexity impose fundamental limits on prediction accuracy."*

Figure 7

It would be better to show signed difference rather than absolute difference here. It is important to know which estimate is hotter! This would be a good place to use the blue-white-red color scale from figure 1.

In addition, there are quite a few additional published estimates that you could compare your model against. Additional comparison datasets include: Shapiro and Ritzwoller (2004); Maule et al., (2005); Purucker et al., (2012); An et al., (2015); Lucazeau, (2019), Haeger et al., (2022); Hazzard and Richards, (2024).

We thank the reviewer for these suggestions. We have revised Figure 7 to show signed differences rather than absolute differences, using a blue-white-red diverging color scale to clearly indicate which estimate predicts higher or lower GHF values relative to our model. We have expanded our comparison to include additional published estimates:

*"To quantitatively assess the relationship between our predictions and existing models, we computed the spatial differences between our GHF map and six published estimates: Fox Maule et al. (2005), Martos et al. (2017), Shen et al. (2020), Lösing and Ebbing (2021), Stål et al. (2022), and Hazzard and Richards (2024)."*

L356: "In instance…"

Should be, "For instance…"

Thanks. Did it!

Figure 8

It appears that many of the observations that you use to validate your model are actually located on the seafloor around Antarctica. While it certainly makes sense to include these data points when so few in situ observations are available, does it really make sense to compare your model against these data when you excluded marine observations from your training data?

Indeed, it is inconsistent to validate our model against marine observations when we excluded marine data from our training dataset. We have removed this validation section from the revised manuscript.

L389-392: "This discrepancy may result from the heterogeneity of local geologic features, differences in raw data processing methods, or the influence of complex processes such as shallow water circulation and unsteady convection in the lithosphere, and further studies are needed to elucidate the underlying mechanisms."

In addition, the discrepancy between your results and those of Shroeder et al. (2014) could be the result of model assumptions made by Schroeder et al. They made very specific and potentially limiting assumptions about the form of the subglacial hydrological system when constructing their inverse model, and those assumptions could potentially introduce errors into their result.

L389-392: We thank the reviewer for this insightful comment regarding the potential influence of model assumptions in Schroeder et al. (2014). We have substantially revised the Discussion section, and this paragraph has been removed as part of the reorganization. Please refer to the revised Discussion for details.

Section 6 Data Availability

This section should be after the Conclusions section, not before it.

Thanks. We have moved the Data Availability section to follow the Conclusions section.

L426: Zenodo link

It would be nice if this link also contained the processed and gridded datasets used as input to your model. While it is true that these datasets are all available at their original sources, it would be nice if it were possible for interested users to access the gridded inputs that you created for your model at one place.

We agree. We have updated the Zenodo repository to include all processed and gridded input datasets used in our model,
in addition to the final GHF predictions and code.

L437: "...which is consistent with the active geological structures." Rephrase this, this sounds awkward. Perhaps try: "...which is consistent with the locations of present-day tectonic and volcanic activity."

We agree that the suggested phrasing is clearer and more precise. We have substantially revised the Discussion section, and this paragraph has been removed as part of the reorganization. Please refer to the revised Discussion for details.

References: The references should be in alphabetical order, not in citation order.

Thank you for pointing this out. We have reorganized the reference list into alphabetical order by first author's surname, following standard formatting conventions.